# Nanoscale redox mapping at the MoS$_2$-liquid interface

He-Yun Du [1,2,3], Yi-Fan Huang[4], Deniz Wong[4], Mao-Feng Tseng [4], Yi-Hsin Lee[1,5], Chen-Hao Wang [5], Cheng-Lan Lin[6], Germar Hoffmann [7,8✉], Kuei-Hsien Chen [1,4✉] & Li-Chyong Chen [1,2✉]

Layered MoS$_2$ is considered as one of the most promising two-dimensional photocatalytic materials for hydrogen evolution and water splitting; however, the electronic structure at the MoS$_2$-liquid interface is so far insufficiently resolved. Measuring and understanding the band offset at the surfaces of MoS$_2$ are crucial for understanding catalytic reactions and to achieve further improvements in performance. Herein, the heterogeneous charge transfer behavior of MoS$_2$ flakes of various layer numbers and sizes is addressed with high spatial resolution in organic solutions using the ferrocene/ferrocenium (Fc/Fc$^+$) redox pair as a probe in near-field scanning electrochemical microscopy, i.e. in close nm probe-sample proximity. Redox mapping reveals an area and layer dependent reactivity for MoS$_2$ with a detailed insight into the local processes as band offset and confinement of the faradaic current obtained. In combination with additional characterization methods, we deduce a band alignment occurring at the liquid-solid interface.

[1] Center for Condensed Matter Sciences, National Taiwan University, Taipei, Taiwan. [2] Center of Atomic Initiative for New Materials, National Taiwan University, Taipei, Taiwan. [3] Department of Chemical Engineering, Ming Chi University of Technology, New Taipei City, Taiwan. [4] Institute of Atomic and Molecular Sciences, Academia Sinica, Taipei, Taiwan. [5] Department of Materials Science and Engineering, National Taiwan University of Science and Technology, Taipei, Taiwan. [6] Department of Chemical and Materials Engineering, Tamkang University, New Taipei City, Taiwan. [7] Department of Physics, National Tsing Hua University, Hsinchu, Taiwan. [8] Center for Quantum Technology, National Tsing Hua University, Hsinchu, Taiwan. ✉email: germar. hoffmann@googlemail.com; chenkh@pub.iams.sinica.edu.tw; chenlc@ntu.edu.tw

Two-dimensional (2D) crystalline materials are consisting of a single layer of atoms and expected to have a significant impact on a large variety of applications[1,2]. For transition metal dichalcogenide (TMDC), monolayers are direct semiconductors, of which a common representative is molybdenum disulfide ($MoS_2$). Monolayer $MoS_2$ has a device relevant direct bandgap of ~2 eV[3] and, in comparison to the bulk, its electron concentration at the surface is enhanced by more than three orders of magnitude[4]. Its stacking variability and the dependence of physical properties with thickness are attractive for property fine-tuning[5]. TMDCs are often combined with other 2D layered materials like graphene or hexagonal boron nitride for device application such as transistors, solar cells, water splitting[6], and sensors. The preparation of respective, high quality material on various substrates is established by means of chemical vapor deposition (CVD). The selection of growth parameters and precursors gives a precise control of its morphology, i.e., flake sizes and layer thicknesses[7].

The well-defined band structure of $MoS_2$ flakes with fixed layer number is ideal to explore the localized electrochemistry of 2D materials as relevant for heterogeneous electron transfer (HET)[8], photoelectrochemistry[9], batteries/capacitors etc. Scanning electrochemical microscopy (SECM) is an established method[10] to locally probe catalytic properties[11] as demonstrated for graphene oxide[12] and $MoS_2$ flakes in feedback mode[13] and biased mode[14]. Whereas the device relevant absolute catalytic activity not only reflects the local catalytic property but also its entanglement[15–18] with the electronic structure, transport properties, geometrical structure, locally, and in interaction with the entire environment and requires a multimethods approach. The electronic structure of pristine ultrathin crystals of $MoS_2$ is intensively discussed in simulation and experiments[19–21] with ambiguities related to growth conditions and measurement environments: the work function by Kelvin probe force microscopy (KPFM)[22–27] and $I$–$V$ curves in devices[28] ($\Phi = 4.49$–5.15 eV), the energetic position of the conduction/valance band edge by scanning tunneling spectroscopy (STS)[29,30] and X-ray photoelectron spectroscopy (XPS)[31,32], the optical band gap by photoluminescence spectroscopy (PL)[21]. The effects of band alignment at interfaces are so far mainly discussed for solid state TMDC-based heterostructures[31,33,34] and semiconductor-liquid interfaces in general[35–38] with few reports on the role of the redox mediator-containing electrolyte on the interfacial band offset in $MoS_2$[14,18].

The research of recent years established thin film $MoS_2$ as a notably potent material for applications with a deep impact of edge and vacancy states[39,40], thickness[41], domain sizes[15,16], electron transfer kinetics[18] etc. on device performances, individually identified, although highly intertwined[8,9,15,16,40,41]. The route towards highly efficient materials is the study and discussion of application relevant properties in conjunction with the full bandwidth of physical properties in different environments and their dependencies among each other[17,18]. This is adressed here for $MoS_2$ and can be equivalently expected to be relevant for 2D materials in general, whenever dimensionality effects modulate electronic and chemical properties.

In AFM-SECM, an atomic force microscopy (AFM)[42] controlled probe approaches the unbiased semiconductor surface (feedback mode), a charge-flow through the local probe is established and gives access to the local electrochemical activity[43]. The electrochemical response current, as detected by the AFM-SECM probe, maps the localized surface electrochemical activity between the 2D material and the liquid interface. Conventionally, SECM is performed in the lift mode, i.e., with the current collecting probe placed at a large distance of typically more than 100 nm above the reactive site (Supplementary Note 1). The lateral resolution depends on the probe radius and the distance from the surface and high-resolution imaging of various substrates is reported for nanoelectrode probes.

Here, we apply in-situ AFM-SECM to layered $MoS_2$ with nanoelectrode probes[42]. With the probe operated in the AFM mode in close nm proximity, combining AFM (with observations on the scale of nm in $x$–$y$ direction and atomic resolution in $z$-direction) and SECM, with a minimum resolution of ~200 nm, to characterize in parallel 2D $MoS_2$ flakes, topological, and electrochemical information in the near field, permits the deconvolution of electronic and topographical information and a more precise evaluation of the electrochemical activity. In combination with KPFM and STM/STS experiments and within the framework of the electron transfer at the solid–liquid interface, the HET behavior in dependence of size and layer numbers of the $MoS_2$ flake-liquid junction with $Fc/Fc^+$ as a redox probe is revealed and discussed. Thereby, the central relevance of the mediator/electrolyte on the band alignment is identified and converted into an experimentally observable photocurrent variation.

## Results

**Imaging the local HET process at the $MoS_2$-liquid junction.** Figure 1a visualizes the experimental approach of the combined AFM-SECM technique and the HET behavior of a representative $MoS_2$-liquid junction with monolayer and bilayer areas present. Conventional, laser assisted AFM monitors the surface morphology on the nm-scale with the sample stored in an electrochemical cell and a redox mediator present. When a bias (oxidation potential) is applied between a distant counter electrode (CE in Fig. 1a) and the probe with an unbiased sample (SECM feedback mode), an anodic probe current is established. The variation of the feedback, i.e., the current after subtraction of the constant, cell geometry related background, with position reflects directly the local electrochemical activity in the vicinity of and activated by the probe apex. Here, the color encoded SECM current map shows, in comparison to the insulating $SiO_2$ surface, a positive (i.e., enhanced) feedback when the probe is on top of the monolayer $MoS_2$ flake as evident for a bipolar electrode. The positive feedback results from the cycling of the oxidant ($Fc^+$ or $DmFc^+$) to the reductant (Fc or DmFc) at the $MoS_2$ flake to a Pt-coated probe, as schematically depicted in Fig. 1b. Thereby, oxidation and reduction of the mediator occur in different regions. Under the probe apex within the recycling area (Region I), $MoS_2$ loses electrons when oxidant (Ox) reduces to reductant (Red) and the probe collects electrons when Red oxidizes to Ox. In the more distant recharging area (Region II), $MoS_2$ receives electrons when Red oxidizes to Ox. Respectively, a charge transfer occurs in the solvent via ions, between the different regions via an electron transfer, and under the probe, a diffusion-controlled cycling of $Fc/Fc^+$ or $DmFc/DmFc^+$ is established.

The rate of mediator regeneration at the substrate governs the extent of the measurable positive feedback as established in previous studies[44–47]. The concentration gradient of $Fc^+$ drives the charge flow between sample and probe[48,49], and of Fc for the recycling. 2D $MoS_2$ flakes perform as a bipolar electrode and provide a significant recycled flux of the bulk redox species toward the probe[45,50]. For n-type semiconductors as $MoS_2$, electrons are the majority charge carrier and migrate as shown to maintain charge neutrality[51]. An accumulation at the space charge layer is created after immersing the semiconductor ($MoS_2$) into the electrolyte solution (Fc) allowing for an increase of electron conductivity at the surface of $MoS_2$. Due to electron accumulation at the $MoS_2$ surface after junction formation and between monolayer $MoS_2$ and a liquid, electrons transfer efficiently from the recharging into the recycling area. The electron transfer behavior between the solid–liquid interface,

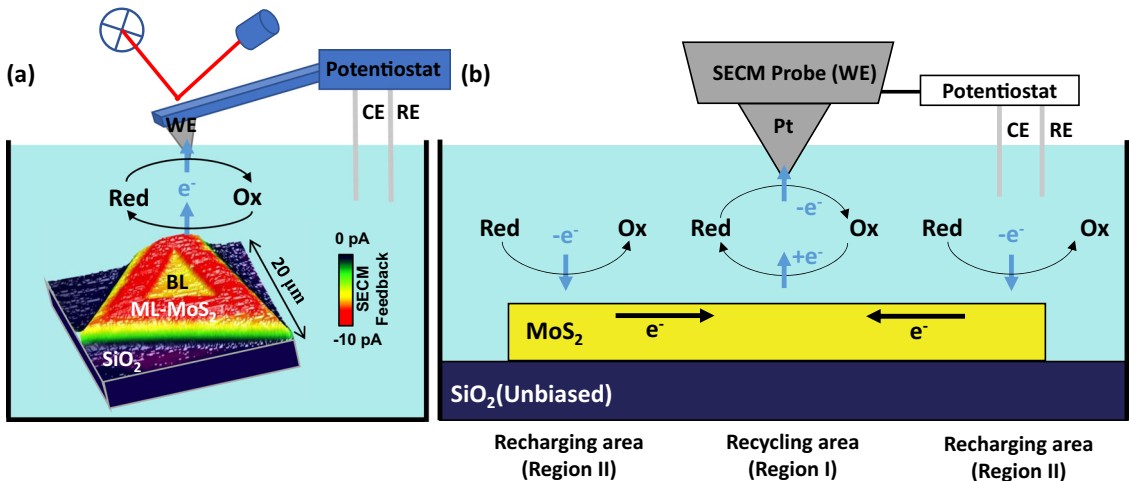

**Fig. 1 Probing the heterogeneous electron transfer at the MoS$_2$-liquid interface via nanoscale redox mapping. a** AFM-SECM setup with the MoS$_2$/SiO$_2$ system, formed of monolayer (ML) and bilayer (BL), immersed in a decamethylferrocene (DmFc) redox mediator electrolyte. For SECM experiments with an unbiased sample, the applied potential at the SECM probe (the working electrode, WE) is controlled by a three-electrode system with a Pt wire as the counter electrode (CE) and a Ag wire the reference electrode (RE). The color of the SECM current map encodes the measured feedback current as flowing through the probe (Scan rate: 20 µm/s, $V_{\text{WE-RE (Ag)}}$: 0.2 V, DmFc mediator). **b** In Region I, recycling of the mediator (curved arrows) from oxidant (Ox) to reductant (Red) carries the feedback with charge neutrality maintained by recharge through Region II. Respectively, electrons flow at the interface (blue arrows) and between the recharging and the recycling area (black arrows). The light blue area represents the redox mediator electrolyte, the yellow area the MoS$_2$ film, and the dark blue area the SiO$_2$ substrate.

such as metal and traditional semiconductor (doped silicon), is established. A similar behavior for MoS$_2$ bulk samples, when the surface is exposed to air, is discussed[4].

**Size effect of MoS$_2$ monolayer in AFM-SECM redox mapping.**
Various high quality MoS$_2$ monolayer flakes were investigated in terms of their topography and electrochemical activity by AFM-SECM in a Fc environment. The impact of the flake size on the feedback was studied. Figure 2a–c present representative AFM images of MoS$_2$ monolayer flakes, having the same height of ~1 nm, as depicted in Fig. 2d, with 3, 20, and 35 µm side lengths. For SECM, the potential difference between the probe and the Ag reference electrode was fixed to 0.6 V for the oxidization of Fc. For each data set, the background current (typically in the range of 100–200 pA) was determined individually on top of the accessible SiO$_2$, which serves as an insulating reference. Figure 2 shows the background normalized SECM feedback maps (e–g) and respective cross-sections (h). Within the noise limit, we find no feedback enhancement for the smallest flake and an increase with size for the two larger MoS$_2$ monolayer flakes (~2 pA and ~6 pA, respectively). The apparent structural resolution obtained in AFM is of the order of 100 nm (Fig. 2d) whereas in SECM of ~2 µm i.e., on the length scale beyond 100 nm, variations in the SECM originate entirely from the local electrochemical activity and are not affected by topographical cross-talk. Considering the probe size for all three presented flakes, the respective recycling areas, which will become apparent when we discuss the bilayer contribution, are much smaller than the recharging areas.

The feedback scales with approximately 8 fA/µm$^2$ (Supplementary Note 2). A fully linear size dependence of the feedback (Supplementary Note 3) can be expected when only the recharge area limits the feedback, with a negligible deviation, i.e., when the sample electron-transport resistance from Region I to Region II or the recycling resistance under the probe are of the same order—which is suggested by the following study. We conclude that here, the observed strong correlation between flake area and feedback is dominantly controlled by the charge flow towards the recharge area (Region II), i.e., the product of ion conductivity and oxidation rate.

**Layer effect of MoS$_2$ bilayer in AFM-SECM redox mapping.**
MoS$_2$ monolayer flakes partially covered with 2nd layer MoS$_2$ were addressed to study the ability of the combined AFM-SECM mode to locally resolve redox active centers with Fc mediator present. Figure 3 shows AFM (a–c), AFM-SECM data (d–f) and feedback cross-sections (g–i) covering 2nd layer islands of (a) µm size and (b) on the 100 nm-scale on top of MoS$_2$ monolayer flakes, with (c) giving a magnified view of (b). The assignment of the specific islands to MoS$_2$ bilayers is verified by standard methods (Supplementary Note 4).

In accordance with the initial observations, we find a feedback enhancement on top of the MoS$_2$ monolayer flakes at a resolution of ~2 µm. In addition, a strong bilayer contrast is detected in SECM maps with a quantitative feedback enhancement of the order of 30% on top of the small and large bilayer islands. The observation of an equal enhancement, i.e., independent of the islands' sizes, suggests that the cycling of the mediator, and respectively, the mapped local electrochemical processes are dominantly confined under the probe. We can respectively deduce a recycling radius to be on the range of ~200 nm, which reflects the obtained resolution at the step between the first and the second monolayer.

The observations of additional enhancement on top of a bilayer within a large monolayer flake and the change in actual resolution require a more detailed analysis. For a qualitative understanding of the observed phenomena, it is sufficient to consider the probe as a point-like redox source located at a distance $d_{\text{probe}}$ above the surface and at $d_{\text{feature}}$ to distant features with altered activity, i.e., the monolayer and the bilayer. All dimensions of the used nonoelectrode probe are on the 0.1 µm (for details see "Methods" section)—the recorded SECM resolution for the monolayer is on the µm range and the given geometry can be discussed in the far-field approximation. The entire monolayer flake contributes to the recharging but not the insulating SiO$_2$—whereas the feedback saturates on sufficiently large monolayers and bilayers, which implies that contributions of a stray current are negligible for the current discussion. Saturation is also the manifestation of the recycling confinement. Therefore, under our experimental conditions the different contributions can be separated. The saturation feedback ratio of $I_{\text{ML}}$ to $I_{\text{BL}}$ is 1:~1.3 and implies an

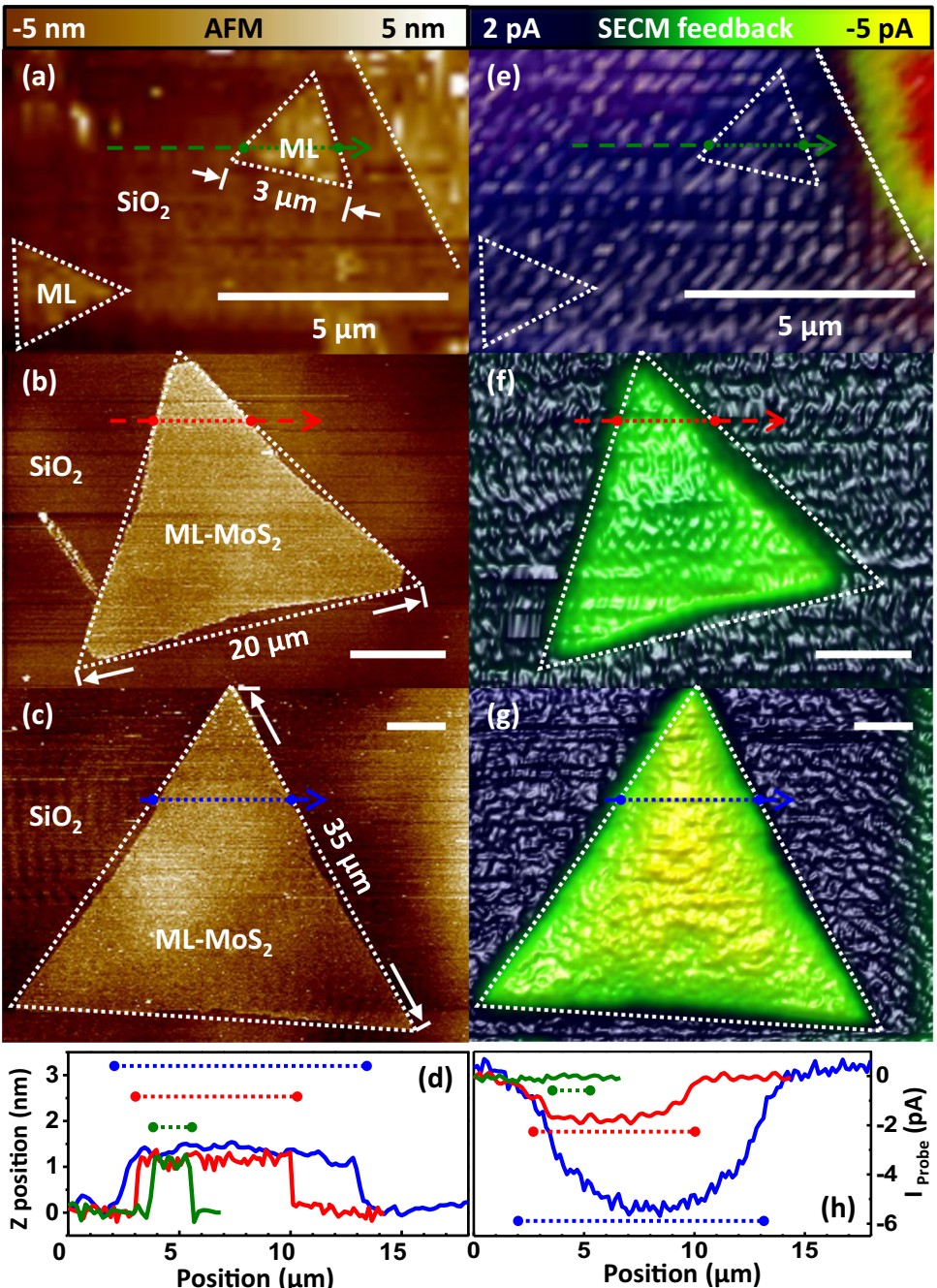

**Fig. 2 Size-dependent AFM-SECM feedback mapping of MoS₂ monolayer flakes.** AFM topographs and SECM feedback maps of marked (dotted lines) MoS₂ monolayer flakes with side length widths of **(a, e)** 3 μm; **(b, f)** 20 μm; **(c, g)** 35 μm. (Scan rate: 20 μm/s, $V_{WE-RE\,(Ag)}$: 0.6 V, Fc mediator). Color and height scales are identical for all images and the white bars indicate a distance of 5 μm. **d** AFM and **h** SECM cross-sectional height profiles along, by respectively color-encoded (green, red, and blue) arrows, indicated paths. The dotted ranges mark the extension of the islands.

increased redox-activity for the bilayer of at least 30% (Supplementary Note 3). For a point-like redox source, the resulting redox concentration gradient can be exactly calculated and respectively, the local point-to-point conductivity at a distance $d$ from the point of closest probe-sample proximity scales with $d^{-3}$ in the far field.

This leaves us with the unexpected finding of altered resolution from the first to the second monolayer. When the probe is located over an insulating surface like SiO₂, at a distant of up to 2 μm from a MoS₂ island (see Fig. 2h) the remaining feedback is driven by the charge flow only towards the more distant, continuously recharged monolayer. Whereas for a

bilayer within a conductive monolayer flake, the resistance scales with distance but is finite and constant towards the monolayer under the probe. A simple resistance network is formed of two parallel paths for discharging towards the probe and a recharging current in series. The feedback signal contributions from distant features are suppressed by the enhancement through the local recycling flux and structural features that are resolved with higher resolution in SECM in close proximity to the bilayer. This is in full agreement with our experimental observations and implies that the observed feedback enhancement ratio only defines a lower value for the increased site activity.

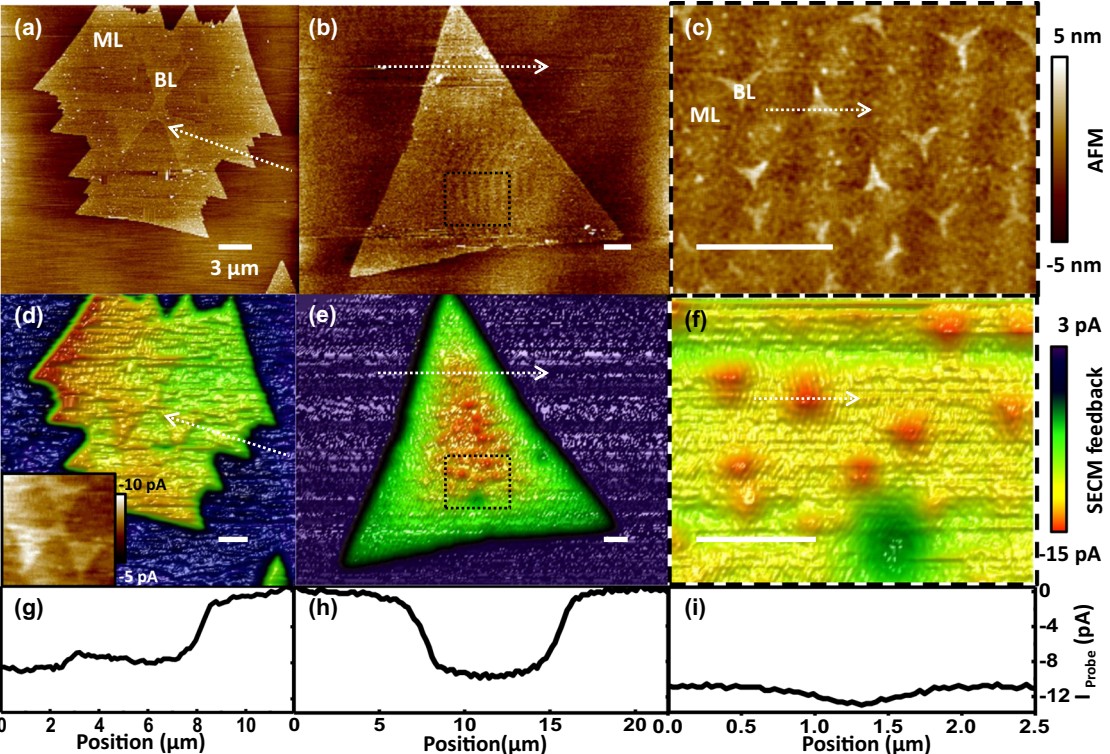

**Fig. 3 SECM feedback mapping of monolayer and bilayer MoS₂.** AFM topographs and SECM maps of (**a**, **d**) three complete pieces of bilayer flakes (**b**, **e**) high density of bilayer flakes on monolayer MoS₂ (**c**, **f**) give a zoomed scan of the area indicated by a dashed box in **b**, **e**. Color and height scales are identical for all images and the white bars indicate a distance of 3 μm. (Scan rate: 20 μm/s, $V_{\text{WE-RE (Ag)}}$: 0.6 V, Fc mediator). The inset of **d** gives a magnified view with adjusted contrast. **g–i** show SECM cross-sectional profiles along the dotted arrows in images **d–f**.

Additionally, observed details are worth-mentioning. We previously assumed a negligible stray current. However, experimental feedback data from monolayer flakes show a slow feedback enhancement towards the flakes' centers (most pronounced in Figs. 2g and 3e). We did not systematically address this effect, with the lateral extension varying between different experiments. We suspect that finite size and probe shape effects might additionally contribute. In Fig. 3f, SECM data reveal a local activity depression (reflected by the green color) which is not correlated to any topographical feature as recorded in AFM. Although we cannot comment on the physical nature, the experimental observation demonstrates the value of SECM to give an important additional access to surface activity beyond simple topographical features. In Fig. 3d, an asymmetry in SECM data of the monolayer is apparent. We verified that misalignment effects in the imaging and normalization processing could be disregarded, though the actual origin is still unclear.

**Layer-dependent work function and band offset.** The observed feedback enhancement for the bilayer is unambiguously related to the increase of activity with Fc as a mediator, as geometrical aspects can be excluded. Therefore, variations of local electronic properties need to be considered. A first indication is given by the variation of the local work function $\Phi$. We performed Kelvin probe force microscopy measurements (see Fig. 4a) in air with the respective probes prior and individually calibrated. We find for the respective work functions $\Phi_{\text{SiO2}} = 5.05$ eV, $\Phi_{\text{ML}} = 5.14 \pm 0.03$ eV, and $\Phi_{\text{BL}} = 5.19 \pm 0.02$ eV. The work function (standard redox potential) of the Fc containing solution is determined to $\Phi_{\text{Fc}} = 5.02$ eV by the cyclic voltammetry method[52], which is shown in Supplementary Fig. 5.

Scanning tunneling microscopy and spectroscopy (STS) experiments were performed for MoS₂ prepared under identical conditions on conductive, highly oriented pyrolytic graphite (HOPG), a necessary change as SiO₂ is insulating. Figure 4c gathers the central findings in STS for MoS₂ of different thickness, i.e., the values for the band gaps and the conduction band onset. Spectroscopic data show that the Fermi energy is below but close to the conduction band, i.e., MoS₂ grown as an n-type semiconductor. The band gaps of $\Delta E^{\text{ML}} = \sim 2.4$ eV for the monolayer and $\Delta E^{\text{BL}} = \sim 2.2$ eV for the bilayer MoS₂ as well as the conduction band onsets by 0.11 eV between the monolayer and the bilayer are reducing with layer thickness. These values are in agreement with previous experiments[23,24,26,27,29,30]. Due to the respective changes of the environments, it can be assumed that absolute values will be different for MoS₂/SiO₂, but that relative changes remain qualitatively valid.

## Discussion

Figure 5a depicts schematically the electronic configuration for each subsystem as experimentally recorded. When the combined system is immersed in the mediator solution (Fig. 5b) under a bias voltage applied between probe and counter electrode, a new equilibrium is established. A charge transfer occurs to compensate for the different work functions and an accumulation region inside the MoS₂ and the Helmholtz double layer/Gouy–Chapman layer is formed. Meanwhile, the applied bias drives a continuous oxidation current.

SiO₂ is an insulating, crystalline semiconductor and the potential difference within the MoS₂ flake is negligible, in comparison to the overall counter electrode-probe resistance, i.e., the applied potential drops over the electrolyte solution. For a 2D sheet (MoS₂) with a local charge source (probe), the resistivity

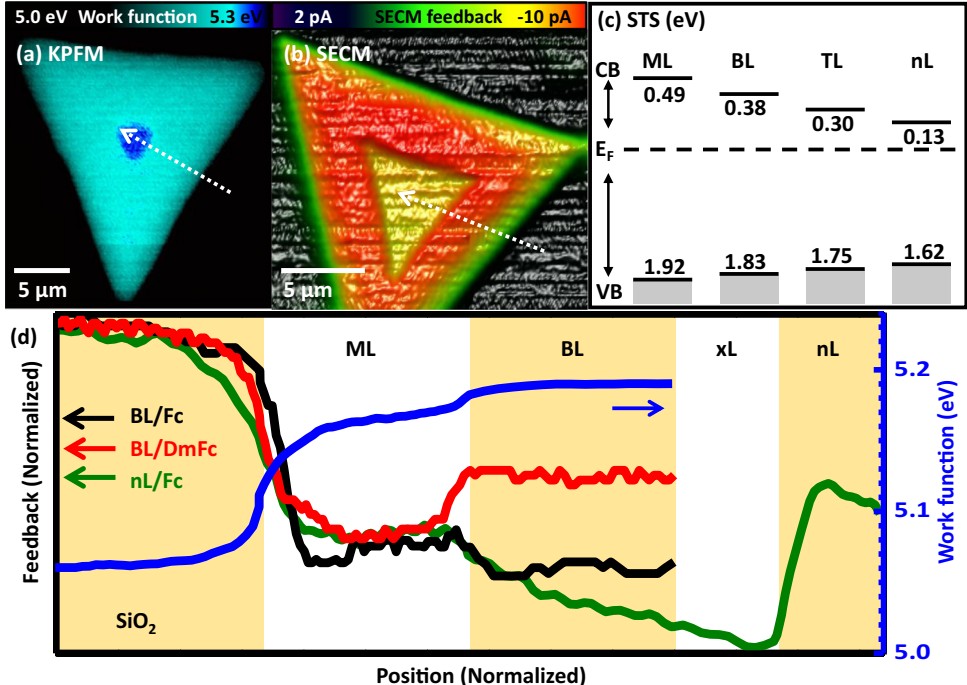

**Fig. 4 Layer dependent physical properties of MoS$_2$. a** Local variation of the work function as recorded in KPFM. (Scan rate: 20 μm/s) and **b** SECM feedback for a bilayer within a monolayer MoS$_2$ flake. (Scan rate: 20 μm/s, $V_{WE-RE (Ag)}$: 0.2 V, DmFc mediator). **c** Band gap and conduction band onset for MoS$_2$ of different thicknesses relative to the Fermi Energy $E_F$ (dashed line) as obtained in STS. (Setpoint: 1.5 V, 49 pA). **d** Combined cross-sectional data from KPFM (blue, from **a**), SECM feedback with Fc mediator (black, from Fig. 3d), with DmFc mediator (red, from **b**) and SECM feedback with Fc mediator of a thick MoS$_2$ flake (green). Data are normalized to the intensity on the monolayer for relative comparison. Data from the thick flake of approx. nL = 8-layers begin on the SiO$_2$ substrate with, in AFM clearly resolved monolayer and bilayer areas, succeeded by a narrow transition (xL) region from second layer to a flat and extended n-layer plateau. Within the transition region, SECM data cannot be attributed to individual layers but are respectively resolution limited. TL = triple layer, CB = conduction band onset, VB = valence band onset. Background color variations indicate change of layer thickness.

scales with one over distance from the center. On the length scale where the resistance contributes significantly, a respective feedback in SECM data can be expected, with no such experimental indication present.

The space charge in a semiconductor junction, here of the MoS$_2$ flake and the electrolyte, which is carried by the work function difference, induces a band offset. The work function difference is by 0.05 eV larger for the bilayer (0.17 eV) in comparison to the monolayer (0.12 eV), and the onset of the conduction band is already lower for the free bilayer system. The electronic configuration during the SECM measurements, i.e., when a bias is applied, is depicted in Fig. 5b. Based on the experimental values albeit the quantitative limitations, a lowering of the bilayer much below the monolayer conduction band onset, and remaining above the Fermi energy can be expected. Respectively, an enhanced charge flow from an energetically lowered conduction band near the Fermi energy into an unoccupied Fc$^+$ state can be deduced. The results of chronoamperometric experiments, i.e., the response of the current flow on an external optical excitation, supports our interpretation of a downward band bending. This is demonstrated in Fig. 5c. A periodic light irradiation results in a parallel, periodic current enhancement. This implies the presence of additional free electrons on the surface. Light excitation creates locally electron-hole pairs and opposite to an upward bending, a downward bending causes an electron accumulation at the interface available for mediator charging as observed and in agreement with our previous interpretation.

In this respect our additional experiments are instructive, which addresses an electronic configuration with a larger charge accumulation on the surface. Figure 4b, d demonstrate a reversed SECM feedback contrast, when a more negative redox potential mediator (DmFc) with a standard redox potential of 0.09 V vs. standard hydrogen electrode ($\Phi_{DmFc} = 4.53$ eV) is used for the otherwise identical sample system. Instead of an additional shift below the Fermi energy, the accumulated charge induces Fermi-level pinning[37,38,53,54] with the conduction band onset fixed to the Fermi energy. Considering that the unoccupied DmFc$^+$ state, which is the final state for the electron transfer, lies close but above the Fermi energy, a potential barrier and a respective feedback reduction emerges. A similar effect can also be expected when Fc is used as a mediator for a sample system with a reduced band gap. Although optimized to grow monolayer and bilayer MoS$_2$ flakes, occasionally thicker films (nL ~ 8 layers) become accessible which have intrinsically a smaller band gap (Fig. 4c). In agreement with a Fermi-level pinning, thick MoS$_2$ layers show the expected reversed contrast (Fig. 4d), in comparison to the monolayer.

Recording the spatial variation of a SECM feedback signal from an unbiased sample immersed in a mediator-containing solution with a recycling of the mediator under a local probe is a common technique in chemistry to study local electrochemical processes. Thereby, the unbiased sample is continuously recharged through a small non-local ion flux over the entire sample. Within the framework of well-established experimental and theoretical work, the process of feedback enhancement due to mediator recycling and limited by diffusion is thoroughly investigated. The feedback enhancement is pronouncedly visible in the sample-probe distance dependence of the feedback and is in general addressed without localization effects due to sample geometry. Localization

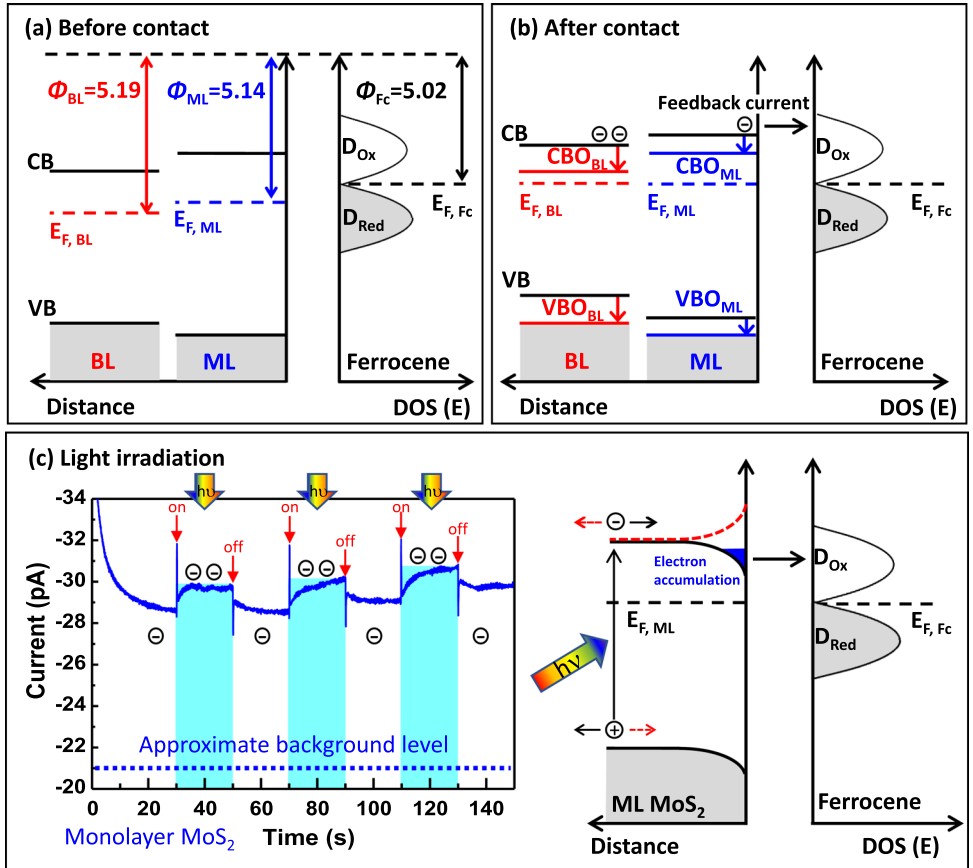

**Fig. 5 Schematic illustration of layer dependent band offsets at the MoS₂-liquid interface. a** Band positions for the mono (blue) and bilayer (red) in MoS₂ and the liquid (black) before contact. Work functions ($\Phi$) are unequal for the subsystems and respectively, their Fermi energies ($E_F$) relative to the vacuum level ($E_{vac}$). During solid–liquid junction formation, the work function differences cause a charge transfer with Fermi-level alignment. After contact (**b**), conduction bands (CB) and valence bands (VB) are respectively offset (CBO and VBO), and the redox potential applied drives an electron current into the unoccupied Fc⁺ states ($D_{Ox}$). $D_{Red}$ are the final occupied states. **c** Chronoamperometric I–t curve shows a light-driven feedback current enhancement with the probe on top of a MoS₂ monolayer flake. At the indicated times (red arrows), a white light source ($h\nu = 0.3-1.8$ eV) is switched on for a period of 20 s, indicated by the bright blue areas. The current enhancement results from interface electron accumulation in response of light irradiation and a downward band bending (black curve), which is not present for opposite bending (red curve). The blue dotted line indicates the approximate, system related background current ($\sim -21$ pA), see for discussion Supplementary Note 6.

effects due to the probe geometry and the respective variation of the diffusion-controlled concentration gradient are a central focus of state-of-the-art simulations[42]. As the sample-probe distance is small in comparison to the recharging area, it is commonly assumed that the feedback signal is carried entirely by the recycling. However, a recent work[45] questioned the full validity of the assumption when the experiments are conducted with the sample-probe distance being much larger than the probe size.

Here, we thoroughly investigate the local SECM signal from unbiased MoS₂ flakes of different thicknesses and sizes immersed in electrolytes with different redox potential mediators. In agreement with a small non-local ion flux for recharging, we find a significant variation of the feedback signal with the size of monolayer MoS₂ on insulating SiO₂. When studying bilayer MoS₂ within a monolayer, size limited MoS₂ flake, the contribution of the non-local recharging flux can be separated from the local flux between probe and sample. Thereby in the case of an Fc mediator-containing solution, the observed enhancement of the feedback of 30% demonstrates that the bilayer is electrochemically more active in the given environment and can be resolved at a resolution down to ~200 nm. The observation of a local feedback enhancement on the bilayer, in combination with a size dependence for the non-local recharging, puts forward that the feedback has to be analyzed and understood within the

framework of a resistance network for the charge flow within the entire experimental setup. Limited by the impact of the exact recharging ion flux, a quantitative enhancement value of the local electrochemical activity cannot be determined without precise knowledge of the resistance values. Further systematic studies might be able to further distinguish between the individual components leading to a refined view.

The observation of varying SECM resolution for monolayers and bilayers, with the resolution significantly larger than the dimensions of the probe, can be intuitively and qualitatively derived from a resistance network formed of the current flow towards the area under the probe in competition towards a distant surface. Moreover, a detectable feedback current is still present at a distance of μm from the flow towards a distant MoS₂ island. With the recharging ion flow towards and the size of the respective monolayer island experimentally known, we can also determine a minimum recycling loss rate of 1% at large distances. As suggested by literature[45], a larger value can be expected.

Finally, a qualitative picture of the observed feedback enhancement of the electrochemical activity for the bilayer in an Fc environment, and the feedback reduction for the bilayer in a more negative redox potential environment formed of DmFc as well as for thicker MoS₂ areas is developed. Supported by additional experimental methods, a lowering of the conduction band onset for the bi-

layers and higher layers can be deduced. The lowering of the conduction band onset increases the amount of available electrons for the reduction process and respectively, the feedback. On the other hand, when the electron density is excessively increased either by a more negative redox potential mediator or by intrinsic sample properties, as in the case of $MoS_2$ multilayers, Fermi-level pinning can be expected. Thereby, the initial electron state in the $MoS_2$ for the reduction process unfavorably ends energetically below the final state in the DmFc and reduces the feedback.

The relation between the electronic structure of carbon related semiconductors and electrolyte interfaces is an important aspect of current research[45,46,51,55]. Light harvesting through enhancement of catalytic activity is offering the potential for carbon-neutral solar fuel production[56]. Current enhancement has been demonstrated in respective micropipette experiments for $MoS_2$[18]. The interpretation within a framework of a spatially varying work function and respective band bending is put forward. Here, a more refined, quantitative view of band bending with layer thickness and support of various methods is given for $MoS_2$.

In summary, experimental evidence is given that localization effects are present. Further resolution enhancement to study local electrochemical processes can be expected based on an inclusion of all, i.e., local and resistivity properties in the design of the experiments. The application of SECM to $MoS_2$ flakes of various sizes and thicknesses in different electrolyte environments reveals a rich physics of band alignment effects, which can be expected for all 2D materials, relevant here to pave a way to enhanced catalytic reactivity.

## Methods

**AFM-SECM**. A Bruker Dimension Icon for AFM with a SECM unit was used to study surfaces in a liquid cell. The SECM measurements were carried out in a PeakForce SECM module, which combines peak force tapping (PFT) imaging mode with the AFM-SECM approach. AFM measurements were performed in PeakForce controlled mode at a force of ~50 nN. PeakForce SECM imaging scans the probe under the main scan first, then lift scan (Supplementary Note 1) with 100 nm above sample surface. SECM measurements in this paper, here explicitly the recording of the feedback current were performed in the main scan mode, i.e., with the probe in close proximity of the sample[42]. The immersed sample was not externally contacted but charge neutrality was given by ion flow within the liquid cell (so called unbiased sample), which results in the measurable net charge flow occurring at the probe, which is by convention called "feedback". The setup of the photoelectrochemical cell used in AFM-SECM system is shown in Supplementary Note 7.

**Electrochemical cell**. The sample was immersed in an electrolyte solution. The electrolyte contained 5 mM Fc (or DmFc) as redox mediator and 0.1 M tetra-butylammonium perchlorate (TBAP) as supporting electrolyte in propylene carbonate. A Pt wire loop was used as a counter electrode and a Ag wire as a pseudo-reference electrode. The standard redox potentials of Fc and DmFc were measured by the nonaqueous $Ag/Ag^+$ reference electrode, then converted to SHE and absolute electrode potential from literature data, which is shown in Supplementary Note 4. Here, we used a Pt nanoelectrode (full diameter of 200 nm and a sharpened tip radius of 25 nm) as the AFM-SECM probe[42]. Electrochemical information of the probes were measured by the cyclic voltammogram and the probe approach curve on top of $SiO_2$ substrate (Supplementary Note 8).

**Feedback current**. The applied potentials between the electrodes cause a current flow within the electrochemical cell, which is partially an ion current within the solvent and an electron current through the sample itself and the probe. By definition, "positive feedback" is called the enhancement of the current due to the local recycling process under the probe with the enlarged charge flow maintained by recharging over the entire conductive surface area. Respectively, "negative feedback" is the current reduction due to shadowing effects of the sample, which interrupts the ion flow. Conventionally, the feedback is normalized against the current observed at a large probe-sample distance. Here, we spatially normalize against the feedback observed on the insulating $SiO_2$, i.e., shadowing effects give a constant offset, and our normalized feedback carries only information on the spatial variation of surface activity for recycling and recharging.

**Normalization of SECM images by cell geometry**. For normalization of the SECM feedback current, the quasi equitemporal background current is mapped line wise on top of $SiO_2$. The probe current on $SiO_2$ (background current) in the

AFM fast scan direction remains (nearly) constant but increases with time in the slow scan direction, as visible in the SECM map—Supplementary Note 9. Supplementary Fig. 9c shows the SECM image after line wise normalization. We find, although the absolute current varies with time, the enhancement remains unaffected which is verified for all experiments presented.

**High quality monolayer and bilayer $MoS_2$ CVD growth**. $MoS_2$ was grown on a Si substrate with 300-nm thick $SiO_2$ by chemical vapor deposition (CVD) method[57]. The schematic diagram of the CVD setup is discussed in Supplementary Note 10. A porcelain boat with sulfur was located in the upstream region outside the furnace. For the growth, we used a sandwich structure formed of the Si substrate with the $SiO_2$ growth surface pointing towards a Mo foil on top of a quartz plate. The sandwich structure was then loaded into a 2-inch diameter quartz tube and centered in the furnace. Before $MoS_2$ growth, the system was flushed with 100 standard cubic center meter (sccm) of argon gas for 10 min. Afterwards, the sample was heated from room temperature to ~800 °C at a rate of 30 °C/min and then maintained at 800 °C for 20 min. Meanwhile, the sulfur (1 g) was heated from room temperature to 160 °C at a rate of 10 °C/min and carried by $Ar/H_2$ gas flow to the growth zone. The Ar and $H_2$ gas flow rates were 50 and 2 sccm, respectively. The operating pressure of CVD was 10 mTorr. The growth with $MoS_2$ flakes on top of $SiO_2/Si$ wafer ended with natural cooling of the system.

**Post-preparation characterization**. After sample preparation, samples were characterized by optical and Raman spectroscopy to verify the morphology in terms of layer thickness (Supplementary Note 4). The Raman related spectra and mapping results were acquired by a confocal Raman spectroscopic system (NTE-GRA Spectra II, NT-MDT spectrum instruments) with a 473 nm excitation laser. The laser power was set at 5.0 mW to avoid possible damage by laser irradiation. The accumulation time at each spectrum spot was 1 s. A 100× objective lens was used to focus the laser and collect the Raman scattered light, and an 1800 lines per mm grating was chosen for spectra acquisition.

In addition, STM and STS experiments, as shown in Supplementary Fig. 11, were performed to verify the relative change of band gaps and band onsets for $MoS_2$ of different thickness. STM and STS measurements were carried out in a homebuilt UHV STM system with the microscope from RHK Technology. The base pressures of the STM chamber and preparation chamber were $6 \times 10^{-10}$ Torr and $2 \times 10^{-9}$ Torr, respectively. After the $MoS_2$/HOPG sample was loaded through the load lock chamber, it was degassed at ~250 °C for 2 h prior to scanning. The mechanically cut PtIr tip was grounded, and a bias voltage applied to the sample.

Amplitude modulated Kelvin probe force microscopy (AM-KPFM) was used to locally map the work function of $MoS_2$ after calibration of the probe work function on HOPG. Thereby, an alternating current voltage of 1.5 V at a frequency of 73 kHz was applied to a Pt-coated/Ir-coated probe. Supplementary Fig. 12 demonstrates experimental results for the monolayer and the bilayer $MoS_2$.

## Data availability

The data that support the findings of this study are available from the corresponding author on reasonable request.

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

## Acknowledgements

This study was financially supported by the Ministry of Science and Technology (MOST) in Taiwan, under the Academic Summit Project (106-2745-M-002-002-ASP and 107-2745-M-002-001-ASP), Science Vanguard Research Project (108-2119-M-002-030), as well as 105-2112-M-007-022-MY3, and 106-2923-M-007-001-MY4. Financial supports by the Deep Decarbonization Project under Academia Sinica (AS-SS-106-02-3 and AS-iMATE-108-31), and the Center of Atomic Initiative for New Materials (AI-Mat), National Taiwan University (107L9008 and 108L9008), Ming Chi University of Technology (VK005-1300-109), along with Center for Quantum Technology, National Tsing Hua University (106N505CE1), from the Featured Areas Research Center Program within the framework of the Higher Education Sprout Project by the Ministry of Education (MOE) in Taiwan, are also acknowledged.

## Author contributions

L.C., K.C., G.H., and H.D. contributed to the conception and design of the experiment. H.D., M.T., and E.L. synthesized the $MoS_2$ sample. H.D., Y.H., and D.W. analyzed the $MoS_2$ sample with OM, Raman and KPFM. H.D., C.W., and C.L. participated in the SECM data discussion. H.D. and G.H. performed STM experiments and wrote the paper. All authors contributed to the discussion and manuscript preparation.

## Competing interests

The authors declare no competing interests.
