## [Peer Review File · Nature Communications]

Reviewers' Comments:

Reviewer #1:

Remarks to the Author:

In the present work, He-Yun et al. study the heterogeneous charge transfer behavior between MoS₂ flakes of various layer numbers and sizes and the liquid interface, using the ferrocene/ferrocenium (Fc/Fc⁺) as redox probe in near-field scanning electrochemical microscopy (SECM). Spatially resolved redox maps reveal a reactivity dependence on the area of the flakes and the number of layers of MoS₂. In combination with scanning tunneling spectroscopy, they finally discuss a band alignment occurring at the liquid-solid interface.

Fundamental knowledge of the electronic structure at the solid-liquid interface in MoS₂ layer is relevant for the understanding and control of the photo- and electrocatalytic behavior of layered MoS₂ regarding, for example, hydrogen evolution and oxygen evolution reactions. The idea of addressing this question using SECM is quite remarkable as it provides parallel topological and electrochemical information. However, it is not clear to me how the results from the work of He-Yun et al. can be translated to the former electrochemical reactions. In my opinion, the paper is more technically oriented to validate the SECM technique in 2D TMDs, but lacks of new insights in the physics at the MoS₂-liquid interface. Therefore, I do not consider this work, as presented, to be published in Nature Communications.

The authors provide a fair description of the observed experimental facts and intuitively argue the space charge in the MoS₂-electrolyte junction -which is carried out by work function difference- to induce a band offset. However, they miss a deeper discussion about the meaning of the observed area and layer dependent reactivity in the monolayer regime, in contrast to the fixed 30% feedback enhancement in bilayer islands. A feedback enhancement vs. flake size comparative plot would help to visualize the correlation between mono- and bilayer flakes, depending on the size range, which is expected to face smaller bilayer flakes.

Hydrogen evolution reaction has been largely related based on the emergence of deep in gap states associated to the presence of S vacancies [Li et al., Nat. Mater. 15, 48–53 (2016)]. More recent studies have identified oxygen substituents as the most abundant point defects in monolayers of TMD semiconductors, which remove the vacancy in gap state [Barja et al., Nat. Comm., 10, 3382 (2019)]. Relevantly, oxide samples show an enhanced catalytic activity for HER [Petô et al., Nat. Chem. 10, 1246–1251 (2018)] compared to fresh ones; and formation of lateral heterointerfaces between defective and pristine regions have also been demonstrated [Kastl et al., ACS Nano 13, 1284, (2019)], with spatial extent similar to the feedback dependence inhomogeneity observed by the authors. The authors need to consider and justify the effect of the former point defects in the observed local variations of the reactivity.

Some more comments are:

1. Units must be included in all graphs.
2. STS parameters and conditions (temperature, environmental conditions, sampling, etc.) should be included in methods.
3. In page 2, line56, it is the authors claim "Electrochemical and topological observations on the scale of nm in x-y direction and atomic resolution in z direction are achieved". Such resolution seems unlikely under the experimental conditions presented in this work. This sentence needs to be justified or removed.
4. Cross-section in Fig. 4(d) is missing in Fig. 4(a) and (b).
5. Figure S1 is missing.

Reviewer #2:

Remarks to the Author:

Summary: The authors studied the relationship between the work function and observed electrochemical reactivity of MoS₂ using several scanning probe techniques (viz, SECM, AFM, STS, STM, and KPFM) with a ferrocene (Fc) redox mediator. Their main results are that multilayers exhibit a higher reactivity than monolayers due to the lower conduction band potential of the multilayers and that bipolar electrochemical activity is proportional to flake size. The data demonstrate clearly a difference in electrochemical reactivity related to the number of MoS₂ layers. However, the analysis and interpretation of the SECM data as well as ambiguous experimental conditions are not convincing. This paper has the potential to be of interest to researchers in a variety of fields, but the paper's contribution to the body of knowledge on TMDCs is not apparent, as the main results of the paper are not discussed in comparison to the literature on TMDCs. The novelty of AFM-SECM (Eifert and Kranz, "Hyphenating Atomic Force Microscopy," *Anal Chem* 2014, 86, 5190-5200) and observation of layer number-dependent electrochemical activity and band gap (Velicky and Toth, "From two-dimensional materials to their heterostructures: An electrochemist's perspective," *Appl Mater Today* 2017, 8, 68-103) of MoS₂ is unclear.

Specific comments and questions:

- Abstract is misleading. Photocatalytic and water splitting properties of MoS₂ should be de-emphasized, as the authors do not present any photocatalytic results or relate their results to the photoactivity of MoS₂. If water splitting is of interest, why was organic solution used? What are "local processes" (line 24)?

- What does "near field" (lines 22 and 54) mean? Do the authors mean small SECM probe-sample distances? Similarly, the description "large distance above the sample" in line 47 is vague. What is a large distance? SECM measurements are done with probe-sample distances less than the probe radius size.

- Instead of the focus on AFM-SECM (see, for example, the paragraph starting on line 53) and achieving high resolution, what may be more interesting in this paper is the ability to compare directly local work function and electrochemical activity at the same location on the same MoS₂ sample. However, this capability is not apparent in the paper as written.

- "Potential Stat" should be "Potentiostat" in Figure 1.

- Several experimental conditions need to be clarified.
 - (a) Supporting electrolyte and solvent need to be verified. Did the authors mean tetrabutylammonium perchlorate (TBAP), which is a more common electrolyte, instead of tetrabutylammonium perruthenate (TBAP) (line 300)? If perruthenate, an oxidizing agent, was used, would it oxidize the redox mediator or MoS₂? Polypropylene carbonate (line 301) is a thermoplastic material and not a liquid solvent.
 - (b) SECM probe dimensions (lines 303-304) are the same as in Ref 20 (Michael et al), and I assume that the authors used similar, commercially available probes. However, cyclic voltammograms (CVs) of the probe in the redox mediator would be helpful as well as an approach curve (i.e., plot of probe current vs. probe-sample distance) of the probe over the Si/SiO₂ surface to confirm the probe dimensions used in this paper. Additional characterization such as SEM would also be helpful.
 - (c) Is the insulating coating on the probes stable in the electrolyte solution?
 - (d) What was probe step size and scan rate in AFM and SECM images? How long did each image take? Can the solvent evaporate within the duration of the measurements?
 - (e) Lift mode in lines 47 and 295-296 as well as how AFM and SECM measurements are performed in this paper need to be better explained. Is SECM probe current measured in the main mode, where the probe is in direct contact with the sample, as shown in Figure S3a (line 297)? In my understanding of PeakForce SECM, two steps are involved: (i) main mode (tapping) is AFM to determine topography and (ii) lift mode is SECM to determine electrochemical reactivity with a

constant probe-sample distance based on AFM profile. How was SECM probe-sample distance of 100 nm achieved and chosen for this paper?

(f) Why was SECM current normalized with respect to negative feedback at Si/SiO₂ instead of current at semi-infinite conditions, as is done typically in SECM measurements, and how was this normalization done? "cell geometry related background" in line 79 needs to be explained.

(g) For Figure S4, experimental details (e.g., sweep rate, electrode area/size, concentration of mediator) for the CVs are missing. I assume that the CVs are for the SECM probe but don't understand the decaying, diffusion-limited current. I would expect steady-state current (i.e., sigmoid shape and independent of potential) for ultramicroelectrodes such as SECM probes.

(h) Experimental details, for example bias current and scan rate for STM data and frequency used for Raman imaging in Figure S6, need to be given. Figure captions in the SI need more detail.

- Fc passivated the probe within a few hours, rendering quantitative comparison between samples difficult (SI section 1). Could impurities in the Fc or solvent absorb on the probe or sample? Did other mediators (e.g., DmFc) exhibit similar, passivating behavior?

- How was the Ag QRE potential determined with respect to the SHE (Figure S4)? Is the potential of the Ag QRE expected to be 0.00 V vs. SHE? Is the QRE potential stable during the measurements?

- The interpretation of the SECM needs to be reevaluated. The statement "the distance limits the maximum achievable resolution and, as resistance for the ion flow increases, the current sensitivity" in lines 49-50 is confusing. SECM probe electrode size determines spatial resolution. I agree with the authors that SECM probe current is affected by probe-sample distance, but this current arises from Faradaic processes (i.e., charge transfer, electron transfer in this case, across the electrode-electrolyte interface) between the probe and redox mediator, not ion flow. For electron transfer to occur, the mediator needs to diffuse to the probe. In negative feedback (such as over an insulator like SiO₂), current decreases at small probe-sample distances because mediator diffusion to the probe is hindered by the sample. In positive feedback (such as over a conductor), current increases due to increased flux of mediator generated at the sample. Commonly, feedback current is analyzed using models for negative and positive feedback (e.g., LeFrou and Cornut, "Analytical Expressions for Quantitative Scanning Electrochemical Microscopy (SECM)," *ChemPhysChem* 2010, 11, 547-556). The authors mention positive/negative feedback in lines 89-92 and 308-316 (although I don't understand why Ref 26 is cited here) but do not apply this analysis to their results. If ion flow does play a role in the SECM current, would current/resistance vary with ionic strength?

- I agree with the authors that resistance in the sample can affect the observed SECM current (e.g., by altering the apparent HET kinetics), but how sample resistance relates to heterogeneous electron transfer (HET) of Fc at MoS₂ needs to be better explained as well as how well the author's model compares with the SECM literature on unbiased samples of finite size (e.g., Ref. 27: Wipf and Bard, *J Electrochem Soc* 1991, Ref 28: Amemiya, *Anal. Chem.* 2007). The discussion of resistance throughout the main text and SI (section S2) is confusing. Is resistance due to the sample or due to electron transfer between the MoS₂ and the electrolyte? What gives rise to this resistance? In line 19 of the SI, where does the contact resistance originate? MoS₂ is electrically isolated from the circuitry (line 304 of the main text). Units need to be added to figure S2. Numbers need to be checked in the SI. On line 17, the area of the reference resistance is (15 mm)² and in the equation on line 43, it is 15 μm². Line 18 has 3 μm² (is it supposed to be 3 μm on a side?).

- In the discussion, results need to be compared to those in literature. How do band gaps, fermi level, and work functions compare with reported values? Can you determine the HET rate constant of Fc and dmFc, and how does it compare with other mediators and reported values?

- In Figure 2e, 3 μm flakes are not observed with SECM, as mentioned in the main text (line 108)

and SI (line 7). Could the flakes detach from the Si/SiO₂? Was AFM/optical imaging done after SECM to confirm that the flakes were still attached?

- On line 115, what area is used to normalize the feedback current (8fA/nm²)?

- Why does the resolution change based on the number of monolayers (see lines 140, 145-146)? I would expect the same resolution for similar SECM parameters and probe size regardless of sample features, as the resolution is determined by the SECM probe radius.

- I disagree with assuming that the SECM probe is a point-like source (line 147). Theoretical models are available for analyzing a conical SECM probe of finite size.

- The following needs to be clarified.

(a) Lines 109-111: The statement beginning "AFM resolution is of the order of 100 nm (Fig. 2(d)), which is image size but not technically limited, in comparison to AFM-SECM resolution of ~2.5 μm. i.e. on the length scale beyond 100 nm."

(b) Lines 119-120: "strong correlation between flake area and feedback is dominantly controlled by the ion flow, i.e. the product of ion conductivity and oxidation rate towards the recharge area (Region II)."

(c) Line 148-149: The statement beginning "All dimensions of the used ultra-microelectrode probe are on the 0.1"

(d) Discussion on lines 154-157, particularly which conductivity is being determined and how d-3 is determined.

- The statements "When the probe is located over an insulating surface like SiO₂, the feedback is driven by the charge flow only towards the more distant, continuously recharged monolayer" (lines 158-159) and "Moreover, a detectable feedback current is still present at a distance of μm from the flow towards a distant MoS₂ island "(lines 272-273) are misleading. The probe reaction (oxidation of Fc to Fc⁺) is driven by potential applied by the potentiostat and not a bipolar reaction with a distant (how distant?) MoS₂ flake. The distant flake is on an insulator (SiO₂) and is unable to deliver charge (electrons) through the sample to the area under the probe. For similar reasons, I disagree with interpreting the paper's results based on Ref. 24 (Amatore et al.). In their experiment, the distant electrode is electrically connected to the sample area under the probe and thus can act like a bipolar electrode.

- Could the enhanced positive feedback current for the center of the larger area flakes in figure 3g (line 167) be due to the large density of bilayer flakes in this region?

- In Figure 3f and line 170, was the unexpected area of decreased current reproducible (i.e., was it observed on subsequent SECM images) and were similar features observed in other samples?

- Could the more negative current on the left side of the flake in figure 3d and lines 172-173 be due to sample tilt or differences in the apparent HET of the sample? Were any adjustments made (experimental or software) to correct sample tilt relative to the probe in all of the SECM images?

- Would it be possible to perform AFM-SECM on MoS₂ on graphite to compare the electrochemical reactivity with STS results? I agree with the authors that the underlying substrate could affect the electronic properties of the MoS₂.

- In figure 5, numbers to vertical axis would be helpful.

- Why are intervals given for some numbers (e.g., the work function) and not others (saturation ratio of 30%)? Are intervals standard deviation or confidence intervals. The intervals for the monolayer and bilayer overlap at 5.17 eV. Are the work functions of the monolayer and bilayer different?

- I disagree with using the term electronegative to describe redox-active species, as electronegativity is an atomic property. When saying more/strongly electronegative, do the authors mean that the mediator is more reducing or has a more negative reduction potential?

Reviewer #3:

Remarks to the Author:

The authors report on the characterisation of heterogeneous charge transfer behaviour of MoS₂ flakes varying the layer numbers and sizes. They applied a combined AFM-SECM approach to probe the reactivity of different number of layers, and mapping the band alignment as well. This manuscript reports original and novel results, after reviewing the manuscript I think this work can be published in Nature Communications after major revisions.

My comments and questions:

- 1) The Introduction and Experimental sections are well written and every step is showed. However, the Introduction is quite straightforward, probably the Authors could have a think to emphasize some points more extending this part a bit.
- 2) The authors chose the Fc and DmFc redox mediators and the propylene carbonate solvent for redox studies. Why the organic solvent is necessary? If the moisture or aqueous contaminants affect, did they apply any drying process for the solvent?
- 3) Both Fc and DmFc experiments showed in the paper, the blank measurements (applying supporting electrolytes only) would worth to be shown and discussed.
- 4) The dopant interaction of several organic solvents is known in the case of graphene and TMDs, have the authors considered this effect, or has any control measurements done using aqueous or other organic media?
- 5) The authors mention that "Due to the respective changes of the environments, it can be assumed that absolute values will be different for MoS₂/SiO₂, but that relative changes remain qualitatively valid." in the "Layer-dependent work function and band offset" part. The effect of substrate (insulator vs. conductor) for electrochemical behaviour of TMDs is also known, so authors should consider this as well, proving with control measurements on both substrates. <https://doi.org/10.1088/2053-1583/aaa4ca>, <https://doi.org/10.1021/acsnano.8b06101>
- 6) The comparison of different number of layers seems to be a bit random, only the ML and BL, or ML and BL and thicker samples are compared in some cases, therefore some summarising explanations/graphs of the systematic study are necessary. More explanation and discussion on the HET dependence of the number of layers are also needed, to underline one of the selling points of the manuscript, the electrochemical reactivity dependence of different number of layers ("heterogeneous charge transfer behavior of MoS₂ flakes of various layer numbers").
- 7) I also reckon that, this work should be publish as a communication in its current form in another more electrochemistry related journal, or the authors should revise and extend it for publication as a full paper in Nature Communications.

Response to Referee Report 1:

We deeply thank Referee 1 for his/her comments and suggestions. We agree that our manuscript has many, more technically oriented aspects, which would certainly qualify it for a respective journal. Explicitly considering Referee Report 2, we see it is mandatory to give sufficient background details so that the reader can clearly identify the differences in comparison to conventional SECM. However, our modification is only a tool – with the previously hidden physics at the interface revealed and discussed.

As also pointed out by the other referees, our approach gives an insight into the present system, which was not previously explored and is benefitting from the chosen mode of SECM operation and the combination with additional techniques. In so far, our approach suggests an application not limited to MoS₂ but to all systems with local electro-chemical processes, as already presented in the discussion section. Nevertheless, to demonstrate a more general value, we performed additional light irradiation experiments (see Fig. 5c), which further supports our interpretation and – as suggested by Referee 3 – we extended the introduction to widen the focus.

Review 1:

(Q1.1) In the present work, He-Yun et al. study the heterogeneous charge transfer behavior between MoS₂ flakes of various layer numbers and sizes and the liquid interface, using the ferrocene/ferrocenium (Fc/Fc⁺) as redox probe in near-field scanning electrochemical microscopy (SECM). Spatially resolved redox maps reveal a reactivity dependence on the area of the flakes and the number of layers of MoS₂. In combination with scanning tunneling spectroscopy, they finally discuss a band alignment occurring at the liquid-solid interface. Fundamental knowledge of the electronic structure at the solid-liquid interface in MoS₂ layer is relevant for the understanding and control of the photo- and electrocatalytic behavior of layered MoS₂ regarding, for example, hydrogen evolution and oxygen evolution reactions. *The idea of addressing this question using SECM is quite remarkable as it provides parallel topological and electrochemical information.* However, it is not clear to me how the results from the work of He-Yun et al. can be translated to the former electrochemical reactions.

In my opinion, the paper is more technically oriented to validate the SECM technique in 2D TMDs, but lacks of new insights in the physics at the MoS₂-liquid interface. Therefore, I do not consider this work, as presented, to be published in Nature Communications.

Answer: See general discussion above. Our research, which was initially indeed aimed for electrochemical reactions, did reveal that first a detailed understanding of

the band alignment is mandatory. Extremely highly cited, the review article of M. Grätzel (Nat. 414, 338 (2001)) identifies band alignment and its understanding as crucial for improvements on photoelectrochemical cells. “The flat band potential is a very useful quantity in photoelectrochemistry as it facilitates location of the energetic position of the valence and conduction band edge of a given semiconductor material. “; “This depends on the applied bias voltage according to the Mott–Schottky equation, where $\Delta\Phi_{SC}$ is the voltage drop in the space-charge layer.” Now we compare the same issue ($\Delta\Phi_{SC}$) by the observation of feedback current difference between ML and BL MoS₂ single crystal, which owing to their band offset difference. Our articles delivers unprecedented access with high spatial resolution into the local processes as band offset and confinement of the faradic current obtained, and unravel the charge transfer behavior at the solid-liquid interface of MoS₂ flakes of various layers and sizes.

(Q2.1) The authors provide a fair description of the observed experimental facts and intuitively argue the space charge in the MoS₂-electrolyte junction -which is carried out by work function difference- to induce a band offset. However, they miss a deeper discussion about the meaning of the observed area and layer dependent reactivity in the monolayer regime, in contrast to the fixed 30% feedback enhancement in bilayer islands. A feedback enhancement vs. flake size comparative plot would help to visualize the correlation between mono- and bilayer flakes, depending on the size range, which is expected to face smaller bilayer flakes.

Answer: This is indeed a very interesting question, which we would love to be able to give an answer – but we are not able with the reasons stated in the manuscript. The feedback enhancement is driven by the recycling area (see Fig. 1), with its size determined to be of the order of (200 nm)². The detection threshold is given by the current resolution in dependence of the size (8 fA/μm²) and would require measurements in the range of Atto-A (10⁻¹⁵ A), which is not feasible. This is demonstrated in Fig. 2a/e with the current vanishing for an island of 100 times the size, when feedback enhancement is expected to be affected.

Fig. 5c gives now even more additional experimental evidence for our intuitive physical interpretation.

(Q3.1) Hydrogen evolution reaction has been largely related based on the emergence of deep in gap states associated to the presence of S vacancies [Li et al., Nat. Mater. 15, 48–53 (2016)]. More recent studies have identified oxygen substituents as the most abundant point defects in monolayers of TMD semiconductors, which remove the vacancy in gap state

[Barja et al., Nat. Comm., 10, 3382 (2019)]. Relevantly, oxide samples show an enhanced catalytic activity for HER [Petô et al., Nat. Chem. 10, 1246–1251 (2018)] compared to fresh ones; and formation of lateral heterointerfaces between defective and pristine regions have also been demonstrated [Kastl et al., ACS Nano 13, 1284, (2019)], with spatial extent similar to the feedback dependence inhomogeneity observed by the authors. The authors need to consider and justify the effect of the former point defects in the observed local variations of the reactivity.

Answer: Vacancies and defects are very interesting aspects additionally contributing to catalytic activity. In the present manuscript, selected data were explicitly chosen from clean samples with a low ~~and over the surface constant~~ abundance of defects and vacancies. In so far, for the study of the principal effects, selected samples offer a constant background variation, unlike the system studied in [Kastl et al., ACS Nano 13, 1284, (2019)]. In Kastl et al. a pronounced rim variation in KPFM can be found (Supplement Figure 3 in above article with add. details) which originates from the specific growth. We also find such rims under altered growth conditions but for the purpose of this manuscript, we excluded such growth studies. S10 clarifies the mentioned issue for the studied films.

Some more comments are:

(Q4.1) 1. Units must be included in all graphs.

Answer: As suggested we added:

Fig. 1: MoS₂ flake size, SECM scan rate and tip applied voltages in caption are added.

Fig. 2: The mediator information, SECM scan rate and tip applied voltages in caption are added.

Fig. 3: The mediator information, SECM scan rate and tip applied voltages in caption are added.

Fig. 4: The mediator information, SECM/KPFM scan rate, SECM/KPFM tip applied voltages and the STS bias/set point in caption are added. In Fig. 4(c), both x axis (flake position) and y axis (flake size) are normalized in order to compare the feedback tendency in different layer number region. In x axis, the SiO₂ monolayer and bilayer side layer are normalized to the same from three different samples (black: bilayer in Fc, red: bilayer in DmFc, green: multilayer in Fc).

Fig.S9. STM/STS parameter added.

(Q5.1) 3. In page 2, line56, it is the authors claim “Electrochemical and topological observations on the scale of nm in x-y direction and atomic resolution in z direction are achieved”. Such resolution seems unlikely under the experimental conditions presented in this work. This sentence needs to be justified or removed.

Answer: The chosen phrasing is indeed (unintentionally) misleading. The stated precision is technically given but observed structures are obviously not on the same scale.

We respectively clarified:

“AFM observations on the scale of nm in x-y direction and atomic resolution in z direction are given, with a min resolution of ~200 nm observed in SECM maps.”

(Q6.1) 4. Cross-section in Fig. 4(d) is missing in Fig. 4(a) and (b).

Answer: Added

(Q7.1) 5. Figure S1 is missing.

Answer: Supplement S1 (as referenced in the manuscript) is the text under S1. For clarity, we added a table S1.

Reviewer 2:

We thank the referee for the intense and knowledgeable questioning, which is often directly or indirectly linked to our interpretational and strongly criticized approach – see Q14.2 “I disagree with assuming that the SECM probe is a point-like source”. We see it therefore mandatory to clarify this issue before discussing detailed aspects, which we thoroughly – and we are optimistic that the referee agrees – handled to his/her fullest satisfaction and certainly improved the readability of the manuscript.

We do fully agree, that a numerical approach for the handling of the (underlying) Poisson equation with all the side conditions (recycling flow etc.) is in general required to properly and reliably interpret any effect related to vertical displacements and structural features under the tip apex.

Our experimental approach, i.e. where we treat/discuss the probe as a point-like source, with the probe operated in closer proximity to the surface than in conventional SECM, relies dominantly on information obtained during horizontal probe displacement with AFM (which is in conventional SECM mode only used after or before but not in parallel) recording the respective vertical displacement. Only because of this available additional information, we are able to identify the crosstalk between proximity related SECM variations and to know the apparent proximity to any surface topography related structure at any time of the SECM recording. Therefore, the SECM contrast observed for the bi-layers at a resolution of ~200 nm defines the lower limit of structural analysis to be performed by numerical methods.

However, we do fully disagree, that in the context of the manuscript, discussion under usage of a point-like source anything more than a minor (if any) variation can be expected from a more complex treatment – and can be traced back to the problem of the E-Field calculation (the gradient drives the mass flow in SECM) between two charged objects with the separating distance significantly larger than object size.

First: The length scale is larger than the range within we can experimentally trace any proximity effects.

Second: The case of SECM is complicated by the mass transport, explicitly of the recycling flow, which is rather inhomogeneous in close sample-probe proximity and in variation with distance. An influence of the vertical separation on our interpretation can be excluded based on the experimental evidence given. An inhomogeneity (which is not

described by our point-like probe approach) can be expected – however, critical is only any change of homogeneity with the experimental conditions, which is here the position. When approaching an edge in a distance much larger than the actual probe-sample distance, where proximity effects do not contribute anymore, the change of inhomogeneity is essentially negligible and respectively the relative recycling rate variation over the entire probe (which again is much smaller than the distance to the edge). The treatment of the given case is in so far equivalent to the classical case of two differently charged objects at a separation much larger than objects dimensions. The limitation is, that we cannot give an interpretation of the absolute values – which we did not do and would require a more complex approach. However, the interpretation of relative changes as we did throughout the entire manuscript – as we chose the experiments and samples - remains unaffected.

(Q1.2) Summary: The authors studied the relationship between the work function and observed electrochemical reactivity of MoS₂ using several scanning probe techniques (viz, SECM, AFM, STS, STM, and KPFM) with a ferrocene (Fc) redox mediator. Their main results are that multilayers exhibit a higher reactivity than monolayers due to the lower conduction band potential of the multilayers and that bipolar electrochemical activity is proportional to flake size. The data demonstrate clearly a difference in electrochemical reactivity related to the number of MoS₂ layers. However, the analysis and interpretation of the SECM data as well as ambiguous experimental conditions are not convincing.

Answer: See the above discussion. In the specific experimental realization with the interpretation basing on the horizontal displacement only (in difference to essential all previous publications) – and vertical effects resolved by the AFM-SECM signal cross-talk – we cannot agree with the reviewer’s statement, that the conditions are not convincing and we do not see any ambiguity, explicitly, as not further specified.

More specifically, although a seemingly rough approximation is used, strong evidence is given by the experimental observations itself, gathered in Fig. 2, that no additional high order terms outside the approximation contribute. Numerically treating the system in a fully 3D approach under the known geometrical parameters (as suggested by the Referee), the feedback value is described by a double integral (sample and probe surface – considering actual ion acceleration along the gradient, it can be simplified to a single integral) of a path integral (ion flow) under consideration of an inhomogeneous distribution of charged and uncharged mediator (cell geometry / boundary conditions). It is then instructive to realize, that in the case of a horizontal movement alone, the mathematical complexity vanishes

for the interpretation exactly in the case, that the feedback reacts on changes in the near field (i.e. approaching step) but does not capture the geometrical properties in the far field. Applied to the results in Fig. 2, we find that the feedback varies in dependence of the distance towards the nearest edge – but is independent of the distance to the corners of the triangularly shaped islands. Mathematically, this implies that only the highest order term contributes, which is given by a point-like probe and all other terms, which come from a geometrically more realistic descriptions of the probe-sample system are negligible. This approximation holds till we experimentally observe deviations, which are given below ~200 nm – from this point onwards, an approach as suggested by the Referee is mandatory and this range is explicitly not included in our discussion – and therefore not considered in the discussion of the lateral extension of BL mapping. Instead, the discussion on the BL is restricted to the case of saturation. i.e. where higher order far-field terms do not contribute to an experimentally observable signal.

This paper has the potential to be of interest to researchers in a variety of fields, but the paper's contribution to the body of knowledge on TMDCs is not apparent, as the main results of the paper are not discussed in comparison to the literature on TMDCs. The novelty of AFM-SECM (Eifert and Kranz, "Hyphenating Atomic Force Microscopy," Anal Chem 2014, 86, 5190-5200) and observation of layer number-dependent electrochemical activity and band gap (Velicky and Toth, "From two-dimensional materials to their heterostructures: An electrochemist's perspective," Appl Mater Today 2017, 8, 68-103) of MoS₂ is unclear.

Answer: We need to emphasize, the value of our work is not to give specific answers to the physics of TMDCs nor to establish an entirely new technique – However, new insight is given, as such, we would not dare to submit to this prestigious journal. Our experiments demonstrate that the evolution of existing techniques and the combination with a broad range of other methods give highly valuable access to a deeper understanding of the fundamental aspects of catalytically active liquid-solid interfaces not previously obtained. TMDCs serves here as an example with new insight obtained – but our approach is not limited to the physics of TMDCs.

Specific comments and questions:

(Q2.2) - Abstract is misleading. Photocatalytic and water splitting properties of MoS₂ should be de-emphasized, as the authors do not present any photocatalytic results or relate their

results to the photoactivity of MoS₂. If water splitting is of interest, why was organic solution used? What are “local processes” (line 24)?

Answer: Although we cannot agree with the reviewer’s comment of de-emphasis, Referee 3 even suggests to enlarge scope of the introduction – as it is a central motivation for the current research, we did tune the localization of potential relevance to less pronounced positions within the abstract and introduction for the current revision.

We specified: “...the local processes as band offset and confinement of the faradic current obtained.”

(Q3.2) - What does “near field” (lines 22 and 54) mean? Do the authors mean small SECM probe-sample distances? Similarly, the description “large distance above the sample” in line 47 is vague. What is a large distance? SECM measurements are done with probe-sample distances less than the probe radius size.

Answer: we specified in the modified version “large distance” with “of typically more than 100 nm” and “near field” with “, with the probe operated in the AFM mode in close nm proximity” – and in the abstract, we added “i.e. in close nm probe-sample proximity”

See above discussion, our point-like probe approach requires the identification where validity can be expected – and respectively, we more precisely defined near and far field / respectively large distance.

(Q4.2) - Instead of the focus on AFM-SECM (see, for example, the paragraph starting on line 53) and achieving high resolution, what may be more interesting in this paper is the ability to compare directly local work function and electrochemical activity at the same location on the same MoS₂ sample. However, this capability is not apparent in the paper as written.

Answer: A very interesting suggestion, we are currently upgrading our system for such a purpose but is currently (and for this paper) not available – and the limitation of the solvent and the mediator on the KPFM measurements needs first to be accurately determined with unknown outcome.

(Q5.2) - “Potential Stat” should be “Potentiostat” in Figure 1.

Answer: Changed as suggested.

- Several experimental conditions need to be clarified.

(Q6.2) (a) Supporting electrolyte and solvent need to be verified. Did the authors mean tetrabutylammonium perchlorate (TBAP), which is a more common electrolyte, instead of tetrabutylammonium perruthenate (TBAP) (line 300)? If perruthenate, an oxidizing agent, was used, would it oxidize the redox mediator or MoS₂? Polypropylene carbonate (line 301) is a thermoplastic material and not a liquid solvent.

Answer: Yes, we agree with the referee that this chemical is miswritten. Therefore, the “tetrabutylammonium perruthenate” is changed to “tetrabutylammonium perchlorate”. Also, the “polypropylene” is changed to “propylene”. The solvent is propylene carbonate indeed. We do apologize for these mistakes.

(Q7.2) (b) SECM probe dimensions (lines 303-304) are the same as in Ref 20 (Michael et al), and I assume that the authors used similar, commercially available probes. However, cyclic voltamograms (CVs) of the probe in the redox mediator would be helpful as well as an approach curve (i.e., plot of probe current vs. probe-sample distance) of the probe over the Si/ SiO₂ surface to confirm the probe dimensions used in this paper. Additional characterization such as SEM would also be helpful.

*Answer: Within the framework of relative changes in the chosen experimental approach and with the confinement area – which reflects directly the actual tip geometry - experimentally determined, the knowledge on the actual tip geometry is redundant. We agree that for the future development, the requested information are interesting for the technically interested readers. We therefore added to the supplement CVs and the approaching curve of the probes as Figure S5 (a) and (b), respectively. The paper by Michael, R. N. et al. on Atomic force microscopy with nanoelectrode tips for high resolution electrochemical, nanoadhesion and nanoelectrical imaging, published in Nanotechnology **28**, 095711 (2017), is cited as Ref. 36 in the main manuscript. As discussed in this paper, AFM-SECM tip is a non-flat sheath nanoelectrode, which is assumed as a semispherical probe with radius of 25nm. In this case, it is not suitable to calculate its probe radius by diffusion-limited current under steady state.*

(Q8.2) (c) Is the insulating coating on the probes stable in the electrolyte solution?

Answer: The stability of the tip is indeed highly relevant. The tip quality decays with time in terms of signal sensitivity. However, as relevant for the present manuscript, the signal enhancement scales proportionally – and the relative signal remains constant (till current saturation terminates running experiments). We added S6 for clarification, see also S1.

See above discussion: we focus on relative changes.

(Q9.2) (d) What was probe step size and scan rate in AFM and SECM images? How long did each image take? Can the solvent evaporate within the duration of the measurements?

Answer: We added respective information to the figures. There is no obvious indication for a change of solvent concentration and respective artefacts observed in our experiments within the experimental timescale.

(Q10.2) (e) Lift mode in lines 47 and 295-296 as well as how AFM and SECM measurements are performed in this paper need to be better explained.

Answer: *The detail measurement and normalization information are added in Fig. S3 and S6. The detail description of lift mode scan is reported in the previous paper:*

Michael, R. N. et al. Atomic force microscopy with nanoelectrode tips for high resolution electrochemical, nanoadhesion and nanoelectrical imaging, Nanotechnology 28, 095711 (2017). – cited as Ref. 37 in the main manuscript.

(Q11.2) Is SECM probe current measured in the main mode, where the probe is in direct contact with the sample, as shown in Figure S3a (line 297)?

Answer: AFM works on the peak force mapping module, which is modified tapping mode instead of contact mode.

(Q12.2) In my understanding of PeakForce SECM, two steps are involved: (i) main mode (tapping) is AFM to determine topography and (ii) lift mode is SECM to determine electrochemical reactivity with a constant probe-sample distance based on AFM profile. How was SECM probe-sample distance of 100 nm achieved and chosen for this paper?

Answer: We found that the feedback current (normalized with background) is reduced with the probe-sample distance within 700 nm (see Fig.S5). We choose

the probe-sample distance around half of the maximum feedback current which is roughly 100 nm as the lift mode scan distance to record. For SECM measurements in this paper, here explicitly the recording of the feedback current was performed in the main scan mode.

(Q13.2) (f) Why was SECM current normalized with respect to negative feedback at Si/ SiO₂ instead of current at semi-infinite conditions, as is done typically in SECM measurements, and how was this normalization done? “cell geometry related background” in line 79 needs to be explained.

Answer: The background current (cell geometry related background) means the current of probe on top of SiO₂ in the same AFM scan line. The probe current on SiO₂ is kept similar in the same AFM scan line. However, the probe current on SiO₂ (background current) will slightly increase with time, as shown in Fig. S6(a). Figure S6 (b) compares the variation of the probe current on SiO₂ surface. The normalization of SECM image is done with respect to the background current since the SECM probe current will reduce the background current (SiO₂ position) in the same scan line. Figure S6(c) shows the SECM image after normalization of probe background current along Y direction.

(Q14.2) (g) For Figure S4, experimental details (e.g., sweep rate, electrode area/size, concentration of mediator) for the CVs are missing.

Answer: We added to the figures as suggested.

(Q15.2) I assume that the CVs are for the SECM probe but don't understand the decaying, diffusion-limited current. I would expect steady-state current (i.e., sigmoid shape and independent of potential) for ultramicroelectrodes such as SECM probes.

Answer: Fig S4 is the CVs of the Pt wire with a diameter around 0.25 mm. The tip CV is attached in Fig. S5a.

(Q16.2) (h) Experimental details, for example bias current and scan rate for STM data and frequency used for Raman imaging in Figure S6, need to be given. Figure captions in the SI need more detail.

Answer: Added to figures as suggested.

(Q17.2) - Fc passivated the probe within a few hours, rendering quantitative comparison between samples difficult (SI section 1). Could impurities in the Fc or solvent adsorb on the probe or sample? Did other mediators (e.g., DmFc) exhibit similar, passivating behavior?

Answer: As discussed above in our reply to question (Q13.2), the probe current is continuously increasing with time both on top of MoS₂ and SiO₂. We observed this phenomenon for both Fc and DmFc in organic solvent. We also observed a current drift in the acid solution but on a significantly larger time scale (days). Although we cannot give a definite answer on potential impact of impurities, we have no indication that additional impurities are present for Fc and DmFc.

We need to stress, that within the framework of the manuscript all analyses are based on relative measurements, as within the same scanline, and different sample systems are probed in SECM and AFM. Therefore, the evolution of the increasing current was monitored during the entire experimental course in parallel for all systems and we verified, that the current drift has no effect on the statements of this manuscript.

(Q18.2) - How was the Ag QRE potential determined with respect to the SHE (Figure S4)? Is the potential of the Ag QRE expected to be 0.00 V vs. SHE? Is the QRE potential stable during the measurements?

Answer: The CV of the Pt wire is measured both with the references of Ag wire and non-aqueous silver/silver chloride reference electrode at the same time in Figure S4(a,b). The Ag QRE potential is determined by comparing with non-aqueous silver/silver chloride reference electrode first, then normalized to SHE as shown in Figure S4(c).

(Q19.2) - The interpretation of the SECM needs to be reevaluated. The statement “the distance limits the maximum achievable resolution and, as resistance for the ion flow increases, the current sensitivity” in lines 49-50 is confusing. (Q19.2a) SECM probe electrode size determines spatial resolution. I agree with the authors that SECM probe current is affected by probe-sample distance, but this current arises from Faradaic processes (i.e., charge transfer, electron transfer in this case, across the electrode-electrolyte interface) **between the probe and redox mediator, not ion flow** (Q19.2b). For electron transfer to occur, the mediator needs to diffuse to the probe. In negative feedback (such as over an insulator like SiO₂), current decreases at small probe-sample distances because mediator diffusion to the probe is hindered by the sample.

In positive feedback (such as over a conductor), current increases due to increased flux of mediator generated at the sample. Commonly, feedback current is analyzed using models for negative and positive feedback (e.g., LeFrou and Cornut, "Analytical Expressions for Quantitative Scanning Electrochemical Microscopy (SECM)," ChemPhysChem 2010, 11, 547-556). The authors mention positive/negative feedback in lines 89-92 and 308-316 (although I don't understand why Ref 26 is cited here) but do not apply this analysis to their results (Q19.2c). (Q19.2d) If ion flow does play a role in the SECM current, would current/resistance vary with ionic strength?

Answer:

(to Q19.2a) The dimensions of the probe are smaller than the probe-sample distance in conventional SECM. The nano-electrode probe is coated with dielectric materials and has an exposed conical Pt tip apex of ~200 nm in height and of ~25 nm in end-tip radius.

(to Q19.2b) Undoubtedly, the current at the liquid-solid interface is Faradaic. We need to admit, that a potential misunderstanding arises from an indeed imprecise wording with no distinction of the dominant contribution (Q19.2d)– as we realized when we tried to identify the source of the obvious misunderstanding.

We respectively changed:

Instead of "The observed size dependence of the feedback of approximately 8 fA/ μm^2 is near-linear with the flake area^{S1}" we changed to: "The feedback scales with approximately 8 fA/ μm^2 (see S1)."

And:

"We conclude that here, the observed strong correlation between flake area and feedback is dominantly controlled by the charge flow (Instead of "Ion Flow") towards the recharge area (Region II) (moved from the end of the sentence), i.e. the product of ion conductivity and oxidation rate."

We thank the referee for identifying this potentially misleading wording.

(to Q19.2c) Negative Feedback is certainly present. The experiments are performed in horizontal displacement only (the impact of vertical displacement is minor and discussed in the manuscript within the framework of confinement effects) and different surface areas present in parallel. Therefore, the relative changes – which are the base of our analysis - are entirely unaffected.

Respectively, we can separate information originating from the geometrical structure and surface properties, not equally accessible in conventional SECM, which is covered by the given references.

(Q20.2) - I agree with the authors that resistance in the sample can affect the observed

SECM current (e.g., by altering the apparent HET kinetics), but how sample resistance relates to heterogeneous electron transfer (HET) of Fc at MoS₂ needs to be better explained as well as how well the author's model compares with the SECM literature on unbiased samples of finite size (e.g., Ref. 27: Wipf and Bard, J Electrochem Soc 1991, Ref 28: Amemiya, Anal. Chem. 2007). The discussion of resistance throughout the main text and SI (section S2) is confusing. (Q20.2a) Is resistance due to the sample or due to electron transfer between the MoS₂ and the electrolyte? (Q20.2b) What gives rise to this resistance? (Q20.2c) In line 19 of the SI, where does the contact resistance originate? MoS₂ is electrically isolated from the circuitry (line 304 of the main text). (Q20.2d) Units need to be added to figure S2. (Q20.2e) Numbers need to be checked in the SI. On line 17, the area of the reference resistance is (15 mm)² and in the equation on line 43, it is 15 μm². Line 18 has 3 μm² (is it supposed to be 3 μm on a side?).

Answer:

(Q20.2a) The answer is already given in Section 2 of SI, S2 (point 2) – the interface.

(Q20.2b) The Faradaic Process

(Q20.2c) We added “(given by the Faradaic process)”

(Q20.2d) X-axis is unitless (relative change), Y-axis “(μm)” added to the flake size.

(Q20.2e) We thank the referee for noticing the (annoying) mistakes, which we corrected (3 μm² is correct).

(Q21.2) - In the discussion, results need to be compared to those in literature. (Q21.2a) How do band gaps, fermi level, and work functions compare with reported values? (Q21.2b) Can you determine the HET rate constant of Fc and dmFc, and how does it compare with other mediators and reported values?

Answer:

(Q21.2a) Band structure values from the literature are added in Table 1. The conditions for sample preparations and experimental conditions in different reports do vary slightly. One has to be cautious, to compare the absolute values from different labs. Therefore, we restrict the data collection to studies with a similar sample preparation and test condition and based on a relative comparison of different layers.

(Q21.2b) Reported HET rate constants are listed in Table 2. It must be noted that the HET rate constant strongly depends on the specific environmental conditions (material, mediator types, solvent types, etc.) and is reported for full surfaces but not for flakes. This also limits the validity of a respective comparison with our

experimental realization. Here, we addressed not full surfaces but flakes of ML or BL MoS₂.

The reported values on all properties show a large variation in publications. Therefore, we decided not to include the given values in the manuscript and conducted no comparison (but relevant papers are fully referenced) as we see no added value to the manuscript.

	ML	BL	Reference	
Band gap	2.11		4	STS
Fermi level	4.50			
Band gap	2.16		5	STS
Fermi level	5.1			
Band gap	2.06	1.89	6	STS
Work function	4.93	4.88	1	KPFM
Work function	4.49	4.59	2	KPFM
Work function	5.15	5.25	3	KPFM

	Diamond k _{app} ^o (cm s ⁻¹)	Pt k _{app} ^o (cm s ⁻¹)	ML-MoS2 k _{app} ^o (cm s ⁻¹)	BL-MoS2 k _{app} ^o (cm s ⁻¹)	Reference
Ferrocene in CH3CN	5.5 ×10 ⁻²				7
Ferrocene in BMIMBF4	5 ×10 ⁻³				7
Ferrocene in PPI3-TFSA		2.3 ×10 ⁻³			7
Hexaamineruthenium(III) chloride in acid			1 ×10 ⁻³	1 ×10 ⁻⁴	8 micropipette

1. Li, F. et al. Layer Dependence and Light Tuning Surface Potential of 2D MoS₂ on Various Substrates. *Small* 13, 1603103 (2017).
2. Ochedowski, O. et al. Effect of contaminations and surface preparation on the work function of single layer MoS₂. *Beilstein Journal of Nanotechnology* 5, 291-297 (2014).
3. Choi, S., Shaolin, Z. & Yang, W. Layer-number-dependent work function of MoS₂ nanoflakes. *Journal of the Korean Physical Society* 64 (2014).
4. Mak, K. F., Lee, C., Hone, J., Shan, J. & Heinz, T. F. Atomically Thin: A New Direct-Gap Semiconductor. *Physical Review Letters* 105, 136805 (2010).
5. Chiu, M.-H. et al. Determination of band alignment in the single-layer MoS₂/WSe₂ heterojunction. *Nature Communications* 6, 7666 (2015).
6. Trainer, D. J. et al. Inter-Layer Coupling Induced Valence Band Edge Shift in Mono- to Few-Layer MoS₂. *Scientific Reports* 7, 40559 (2017).

7. *Heterogeneous electron-transfer rate constants for ferrocene and ferrocene carboxylic acid at boron-doped diamond electrodes in a room temperature ionic liquid. Electrochimica Acta 94, 49-56 (2013).*

8. *Velický, M. et al. Photoelectrochemistry of Pristine Mono- and Few-Layer MoS₂. Nano Letters 16, 2023-2032 (2016).*

(Q22.2) - In Figure 2e, 3 μm flakes are not observed with SECM, as mentioned in the main text (line 108) and SI (line 7). Could the flakes detach from the Si/ SiO₂? Was AFM/optical imaging done after SECM to confirm that the flakes were still attached?

Answer: See above discussion on AFM in parallel to SECM. We can disregard detachment of the flakes. SECM is measured in parallel to AFM. Flakes appear in AFM over many individual lines and a detachment would be unambiguously discernible.

(Q23.2) - On line 115, what area is used to normalize the feedback current (8fA/nm²)?

Answer: (See also answer 5 to Referee 1. We apologize for the typo of the unit nm² vs. μm^2). Thereby, the newly added table S1 reveals, how we deduced the given values.

(Note, we changed the order of the following two questions as answers build upon each other)

(Q24.2) - I disagree with assuming that the SECM probe is a point-like source (line 147). Theoretical models are available for analyzing a conical SECM probe of finite size.

Answer: Here, we fully disagree with the referee – see discussion above – the point-like source approach is only applied when we experimentally cannot detect any higher-order contributions, which then would indeed require a more detailed 3D numerical modelling. This is here not given.

(Q25.2) Why does the resolution change based on the number of monolayers (see lines 140, 145-146)? I would expect the same resolution for similar SECM parameters and probe size regardless of sample features, as the resolution is determined by the SECM probe radius.

Answer: It is a seemingly surprising experimental finding that the resolution depends on layer number and did initially puzzle us as well (for the very same

reasons mentioned by the Referee). More advanced numerical treatment of the geometry did not reveal any respective contribution. Our experimental approach (see discussion above), allows for an unambiguous conclusion that crosstalk and confinement effects can be disregarded – in difference to conventional SECM.

We think, that the manuscript already gives a clear answer, which is in short the following:

Whereas handling the charging and recharging – as described in the manuscript – as two independent resistance networks immediately reveals the origin of different distance dependences. When approaching the monolayer edge from SiO₂, the hampered recharging of the insulating SiO₂ causes a redirection of the current flow to the more apart and efficiently recharged monolayer whereas when approaching the 2nd layer, efficient recharging of the monolayer underneath is maintained by the conductive layer. Therefore, the enhancement current related to the 2nd layer appears only at closer vicinity.

(Q26.2) - The following needs to be clarified.

(a) Lines 109-111: The statement beginning “AFM resolution is of the order of 100 nm (Fig. 2(d)), which is image size but not technically limited, in comparison to AFM-SECM resolution of ~2.5 μm. i.e. on the length scale beyond 100 nm.”

*Answer: We modified the manuscript for clarity: “**The apparent structural resolution obtained in AFM** is of the order of 100 nm (Fig. 2(d)) whereas in SECM of ~2.5 μm. i.e. on the length scale beyond 100 nm, therefore, variations in the SECM originate entirely from the local electro-chemical activity and are not affected by topographical cross-talk.”*

It is worth noting that within the context of the referee’s criticism on different points, that the observation of a cross-talk decoupling beyond 100 nm, a strong confinement can be deduced and respectively specifies the range within the validity of a point-like approach can be falsified and beyond which a point-like probe approach is reasonable, i.e. in our discussion when the distance (and variations) are on a length scale much beyond 100 nm. Otherwise, as demonstrated in Fig. 3f of the main manuscript, 2nd ML islands would be traceable at larger horizontal distances.

(Q27.2) (b) Lines 119-120: “strong correlation between flake area and feedback is dominantly controlled by the ion flow, i.e. the product of ion conductivity and oxidation rate towards the recharge area (Region II).”

Answer: See Q19.2b, the sentence is unintentionally misleading and respectively corrected.

(Q28.2) (c) Line 148-149: The statement beginning “All dimensions of the used ultra-microelectrode probe are on the 0.1”

Answer: The unit appeared at a later position. This is indeed potentially confusing. We changed:

0.1 μm range – the recorded SECM resolution for the monolayer is on the μm

(Q29.2) (d) Discussion on lines 154-157, particularly which conductivity is being determined and how d-3 is determined.

Answer: As we refer to the “redox concentration gradient”, the charge conductivity due to ion flow can be unambiguously identified. In the case of large separations of a (respectively small – considering the dimensions of the one used) probe from a surface, the potential gradient (and of the concentration gradient respectively) scales with d-2. Considering the additionally extended diffusion path (scales with d) from probe to sample results in the above statement.

(Q30.2) - The statements “When the probe is located over an insulating surface like SiO₂, the feedback is driven by the charge flow only towards the more distant, continuously recharged monolayer” (lines 158-159) and “Moreover, a detectable feedback current is still present at a distance of μm from the flow towards a distant MoS₂ island “(lines 272-273) are misleading. The probe reaction (oxidation of Fc to Fc⁺) is driven by potential applied by the potentiostat and not a bipolar reaction with a distant (how distant?) MoS₂ flake. The distant flake is on an insulator (SiO₂) and is unable to deliver charge (electrons) through the sample to the area under the probe. For similar reasons, I disagree with interpreting the paper’s results based on Ref. 24 (Amatore et al.). In their experiment, the distant electrode is electrically connected to the sample area under the probe and thus can act like a bipolar electrode.

Answer: Here is an obvious misunderstanding. We clearly speak of the feedback current from the probe to the continuously recharged monolayer (btw (how distant? – is answered by the experimental data presented in the figure 3 and the respective manuscript statement: “MoS₂ monolayer flakes at a resolution of $\sim 2 \mu\text{m}$ ”).

(Q32.2) - Could the enhanced positive feedback current for the center of the larger area flakes in figure 3g (line 167) be due to the large density of bilayer flakes in this region?

Answer: We considered this possibility for the original manuscript submission and we think we can now disregard this possibility. Such an effect should have a similar artefact emerging in 3d/e (and other data), which is not observed.

(Q33.2) - In Figure 3f and line 170, was the unexpected area of decreased current reproducible (i.e., was it observed on subsequent SECM images) and were similar features observed in other samples?

Answer: We find it important to include alternative effects, which are beyond our explanation if present, instead of selecting those images we like. Of course we verified that it is not an artefact – already the feature in 3f is a reproduction of the same effect in the enlarged view scanned

We remain with our original statement: “In Fig. 3(f), SECM data reveal a local activity depression (reflected by the green color) which is not correlated to any topographical feature as recorded in AFM. Although we cannot comment on the physical nature, the experimental observation demonstrates the value of SECM to give an important additional access to surface activity beyond simple topographical features.”

(Q34.2) - Could the more negative current on the left side of the flake in figure 3d and lines 172-173 be due to sample tilt or differences in the apparent HET of the sample? Were any adjustments made (experimental or software) to correct sample tilt relative to the probe in all of the SECM images?

Answer: We refer to the statement on our original manuscript: “In Fig. 3(d), an asymmetry in SECM data of the monolayer is apparent. We verified that misalignment effects in the imaging and normalization processing could be disregarded, though the actual origin is still unclear.”

Note: As AFM and SECM are conducted in parallel an artificial wrong tilting for constant height measurements as in conventional SECM is not possible here.

(Q35.2) - Would it be possible to perform AFM-SECM on MoS₂ on graphite to compare the electrochemical reactivity with STS results? I agree with the authors that the underlying substrate could affect the electronic properties of the MoS₂.

Answer: The combination of all experiments within one setup and conducted in parallel is the dream of all experimentalists. However, the presence of the solvent is problematic for STS and KPFM. In SECM large islands on the 10 μm range are preferred – in STM on the 100 nm to 1 μm scale, so that under identical conditions different layers can be studied to minimize systematic contributions of altered tips etc. The combination would be ideal if feasible – instead, we focused on comparative studies under optimized conditions related to the used techniques and consider the impact of sample variations in the analysis.

(Q36.2) - In figure 5, numbers to vertical axis would be helpful.

Answer: Fig. 5 is, as clearly stated in the caption, a schematic illustration. Values are given in the text, with all relevant background information so that readers can evaluate each individual parameter in the context of all relevant technical information. A visually sufficiently focused gathering of information from many different sources can therefore not be achieved.

(Q37.2) - Why are intervals given for some numbers (e.g., the work function) and not others (saturation ratio of 30%)? Are intervals standard deviation or confidence intervals. The intervals for the monolayer and bilayer overlap at 5.17 eV. Are the work functions of the monolayer and bilayer different?

Answer: We answer in reverse order. The work functions for ML and BL are different. The stated error values reflect statistical error (which is minor as seen from Fig. 4d) and systematic error from repetitive realization of the measurement (different tips, sample, solvents). Thereby, the relative change is not affected by systematic errors – but as demonstrated in the same figure, the relative change is beyond the statistical error.

The seemingly precise saturation ratio was not intended, but a meaningful error can also not be stated (as the exact exp conditions do play a role which we can not individually separate). We respectively changed at different positions within the manuscript to avoid the impression of an unintended accuracy (for example “...enhancement of the order of 30 % on ...”).

(Q38.2) I disagree with using the term electronegative to describe redox-active species, as electronegativity is an atomic property. When saying more/strongly electronegative, do the

authors mean that the mediator is more reducing or has a more negative reduction potential?

Answer: We agree, the wording is not appropriate. We mean more negative reduction potential using normal hydrogen electrode (NHE) as a reference.

Reviewer 3:

With thank the Referee for his/her valuable comments and we follow his/her suggestion to widen the scope of the introduction and respectively, widen the potential application of our finding by introducing with Fig. 5c new data on light irradiation experiments.

The authors report on the characterization of heterogeneous charge transfer behavior of MoS₂ flakes varying the layer numbers and sizes. They applied a combined AFM-SECM approach to probe the reactivity of different number of layers, and mapping the band alignment as well. **This manuscript reports original and novel results**, after reviewing the manuscript I think this work can be published in Nature Communications after major revisions.

My comments and questions:

(Q1.3) The Introduction and Experimental sections are well written and every step is showed. However, the Introduction is quite straightforward, probably the Authors could have a think to emphasize some points more extending this part a bit.

Answer: In order to emphasize the importance of semiconductor properties of 2D TMDCs in the introduction part, we further elaborated the current status of knowledge.

The electronic structure of pristine ultrathin crystals of MoS₂ is intensively discussed in simulations and experiments. The work function is accessible by Kelvin probe force microscopy (KPFM) and I-V curves in devices with strong variations ranging from 4.49-5.15 eV depending on growth conditions and the measurement environment, the energetic position of the conduction/valance band edge by scanning tunneling spectroscopy (STS) and X-ray photoelectron spectroscopy (XPS), the optical band gap by photoluminescence spectroscopy (PL).

The effects of band alignment at interfaces is so far mainly discussed for solid state TMDC based heterostructures junctions and semiconductor/liquid interfaces in general. Scanning electrochemical microscopy (SECM), also used here, is an established method to locally probe catalytic properties as demonstrated for graphene oxide and MoS₂ flakes in feedback mode and biased mode.

Fig.5(c) is added to show the feedback current enhancement of MoS₂ in response of light irradiation experiments.

(Q2.3) The authors chose the Fc and DmFc redox mediators and the propylene carbonate solvent for redox studies. Why the organic solvent is necessary? If the moisture or aqueous contaminants affect, did they apply any drying process for the solvent?

Answer: see also answer to (Q22.2): in short, in aqueous solution, flakes are unstable. Most important parameter here is the energy level (redox potential) of the mediator in the liquid. There are similar mediators in the aqua solution like ferrocenemethanol or potassium ferricyanide. The problem here is that MoS₂ flakes on SiO₂ wafer will peel off due to the hydrophobicity difference between the MoS₂ flake and SiO₂.

In our observations, the organic solvent is quite stable during the experiment.

(Q3.3) Both Fc and DmFc experiments showed in the paper, the blank measurements (applying supporting electrolytes only) would worth to be shown and discussed.

Answer: There is no feedback current of MoS₂ flake observed when we applied the supporting electrolytes only.

(Q4.3) The dopant interaction of several organic solvents is known in the case of graphene and TMDs, have the authors considered this effect, or has any control measurements done using aqueous or other organic media?

Answer: We did not observe the doping effect during the SECM measurement. The acquired time of one SECM image is usually around 10-20 mins. Normally the feedback current (normalized by the SiO₂ current at the same time) is not changed within one hour. We will try to understand this fundamental dopant effect in our following experiments.

(Q5.3) The authors mention that “Due to the respective changes of the environments, it can be assumed that absolute values will be different for MoS₂/ SiO₂, but that relative changes remain qualitatively valid.” in the “Layer-dependent work function and band offset” part. The effect of substrate (insulator vs. conductor) for electrochemical behavior of TMDs is also known, so authors should consider this as well, proving with control measurements on both substrates.

Answer: We agree, each subsystem of the Substrate/MoS₂/Mediator system will impact the band structure and the electrochemical behavior and goal for the future

should be to explore the specific aspects. In this manuscript and beyond the coherence of results of different methods with 5c added, we show the validity of our interpretation for a change of the mediator. For the benefit of a clear result, the change from an insulating to a conducting substrate will also affect the recharge flow through the substrate in a non-trivial way. Although we fully agree that the Referee suggestion is great and in line with our thoughts, it also covers significant experimental barriers.

(Q6.3) The comparison of different number of layers seems to be a bit random, only the ML and BL, or ML and BL and thicker samples are compared in some cases, therefore some summarizing explanations/graphs of the systematic study are necessary. More explanation and discussion on the HET dependence of the number of layers are also needed, to underline one of the selling points of the manuscript, the electrochemical reactivity dependence of different number of layers (“heterogeneous charge transfer behavior of MoS₂ flakes of various layer numbers”).

Answer: Sample preparation was by purpose optimized for mono- and bi-layer growth, with both systems systematically addressed. Additional data for higher layer numbers are used only if available.

Table 1 is added to compare the feedback, band offset and layer number. Also, Figure 5 (c) explained the increased enhanced feedback current due to the photo-excitation to generate electrons and holes. This feedback current enhancement also implied the n type MoS₂ semiconductor is bending downward after immersing in electrolyte.

(Q7.3) I also reckon that, this work should be publish as a communication in its current form in another more electrochemistry related journal, or the authors should revise and extend it for publication as a full paper in Nature Communications.

Answer: We thank the referee for his/her general comments in the beginning before itemized questions, that “This manuscript reports original and novel results, after reviewing the manuscript I think this work can be published in Nature Communications after major revisions.” We thank all the reviewers’ suggestions, the quality of this revised manuscript is greatly improved.

Reviewers' Comments:

Reviewer #1:

Remarks to the Author:

The authors have fairly discussed mayor concerns addressed in the referee's reports and carried out additional measurements to support their interpretation. I think this work can be published now in Nature Communications.

Reviewer #2:

Remarks to the Author:

I thank He-Yun Du et al. for their responses to my questions and updating their manuscript entitled "Nanoscale redox mapping at the MoS₂-liquid interface," particularly in clarifying faradic, ion, and charge flow processes and why they modeled the SECM probe as a point source. I understand better their experimental setup and analysis. However, I still question the novelty of the work as it relates to spatially resolved measurements of mono-, bi-, and multilayer MoS₂ or other 2D materials, even at the nanoscale. In fact, the authors cite a paper (Ref. 36: MR Nellist et al. *Nanotechnology* 28, 095711 (2017)) demonstrating nanoscale redox mapping of 2D materials with the same commercial system (Bruker PeakForce AFM-SECM) used in the manuscript, and the sentence "The nanoelectrode probe is coated with dielectric materials and has an exposed conical Pt tip apex of ~200 nm in height and of ~25 nm in end-tip radius." (line 62) is word-for-word the same as that in the abstract of that reference. In their rebuttal, the authors mention "the evolution of existing techniques and the combination with a broad range of other methods" and that "TMDCs serves here as an example with new insight obtained – but our approach is not limited to the physics of TMDCs," but the manuscript, as written, does not clearly describe in more detail how these insights can be expanded to other applications or materials. Given the body of existing work on MoS₂ electrochemistry and band structure (e.g., Choi, S et al. 2014; Trainer, D. J. et al. 2017; Velický, M. et al. 2016; Velicky and Toth 2017 cited in the manuscript) and the focus on demonstrating the use of scanning probe systems to characterize MoS₂, this manuscript may be better suited for a more technically oriented journal.

These are the specific comments and concerns I have after reading the authors' rebuttal:

(1) I appreciate the authors' diligence in reading and summarizing literature on heterogeneous electron transfer (HET) rates of Fc and agree with their caution in comparing values obtained under different experimental conditions (Q21.2). However, I think that putting their results (how apparent HET and band gap relate to the number of layers and flake size) in context with reported values is important.

(2) Although I appreciate the addition of the photoelectrochemical measurements (Fig. 5), I don't see a clear correlation between the photoelectrochemical water-splitting at TMDCs mentioned throughout the manuscript and the measurements performed in organic solvents. I would have liked to see an explanation as to why organic supporting electrolyte was used (e.g., organic electrolytes were used instead of aqueous ones due to delamination of the MoS₂ flakes). I also would have liked to know how the band alignment in organic solvent would compare with that in aqueous solution (see for example, LF Schneerneyer and MS Wrighton "Flat-Band Potential of n-Type Semiconducting Molybdenum Disulfide by Cyclic Voltammetry of Two-Electron Reductants: Interface Energetics and the Sustained Photooxidation of Chloride" *J. Am. Chem. Soc.* 1979, 101, 6496)

(3) I appreciate the authors clarifying the electrochemical measurements by adding more details to the figure captions as well as correcting units and electrochemical details (e.g., supporting electrolyte). However, I stand by my original assessment that the interpretation and discussion of the electrochemical results are ambiguous:

(a) Regarding Q18.2, I still don't understand why the potential of Fc versus the SHE (standard

hydrogen electrode) is the same that versus the Ag quasi-reference electrodes in Fig. S4c and how the authors arrived at their values. I would expect these values to be different, although I agree that 0 V vs. SHE is comparable to -4.44 eV for vacuum. The authors mention using both the (SHE) and Ag quasi-reference electrode (lines 100 and 107 in supporting information) as well as a "nonaqueous" silver-silver chloride reference in their response. The SHE is difficult to use in practice, and I wonder if they used a silver-silver ion electrode, which is more commonly used in organic electrolytes, instead of silver-silver chloride.

(b) In figure 5, I'm not sure why the current for the "off" photoelectrochemical measurements (280 to 300 pA) is much higher than that in the SECM images (ca. 1-15 pA). I would expect these values to be similar if the probe-MoS₂ distance is similar for all of the measurements.

(c) The authors answered my question about using the word electronegativity to describe the reducing potential of redox-active species (e.g., lines 276 and 295) in their written response (Q38.2) but did not update the corresponding manuscript text.

(4) I appreciate the authors including CVs and approach curves of the SECM probe in Figure S5 (Q7.2). These additions help me better understand the experimental conditions. However, the probe was operated at 0.6 V, where the current is still changing (in other words, it is potential-dependent), and the current value is obviously not stable or reproducible at 0.6 V, as the probe current at 0.6 V was different for all three probes. I would have expected the probe to be poised at a potential (e.g., 0.7 V) where the current is at steady state (independent of potential) as well as more consistent between probes so that any change in current can be attributed to a change in redox activity of the substrate and not due to factors such as a drift in potential of the silver quasi-reference electrode.

(5) For Q30.2, I am still unsure what a distant MoS₂ island is. If the island is another, isolated MoS₂ flake on Si/SiO₂ (an insulator), then I don't understand how charge (electrons) are being passed through an insulator between the two flakes. If the island refers to two different areas on the same flake, then I can understand how the MoS₂ (a semiconductor) is able to act as a bipolar electrode.

(6) For Q35.2, the authors may have misunderstood my question. I am not asking if they could combine all the measurement probes into one instrument but rather if they could do different measurements on the same type of sample, namely, measure MoS₂ on graphite (a conductive substrate) using the AFM-SECM and comparing those results to AFM-SECM imaging on MoS₂ on silicon oxide (an insulator) to see if the substrate (e.g., conductor vs. insulator) affects the observed electrochemical reactivity of the MoS₂.

Reviewer #3:

Remarks to the Author:

Comments to the Authors regarding to the resubmitted version of their revised version of "Nanoscale redox mapping at the MoS₂-liquid interface" (226060_1_art_file_5029456_qnn0gd)

The authors revised their original work, and resubmitted an improved version, which attempts to solve the three Reviewers' comments.

In my case, I see an improvement, and think that the manuscript had revised well, although it would improve more anytime. Regarding to my comments and question for the experimental and results parts are addressed, I am glad to see this. While one of my comment concerning the fundamental aspects of the current work is remained untouched, namely the substrate effect. I can understand that in the current COVID-19 situation, we should focus current results and utilise those as much as possible without trying to extend the existing content applying further experiments. But the Introduction part is still confusing and contains missing parts, pieces of works for me. Please, consider to revise more on the Introduction part to try to emphasize more

the place and the role of your approach (the in-situ atomic force microscopy assisted SECM system). I mean we have knowledge of EC/PEC properties on MoS₂ and other TMDCs varying layer numbers, defect density, doping agent, surface treatments (passivation/activation), etc. Several research groups obtained these works using SECM, SECCM, and a microdroplet-based EC/PEC technique. I added some doi numbers of these papers below, supporting the Authors to find a way to emphasize the position of their work in the current view.

<https://doi.org/10.1002/celc.201500103>

<https://doi.org/10.1039/C5EE02530C>

<https://doi.org/10.1039/C5CP02490K>

<https://doi.org/10.1021/acs.nanolett.5b05317>

<https://doi.org/10.1039/C7SC02545A>

<https://doi.org/10.1021/acs.jpcc.7b12715>

<https://doi.org/10.1021/acs.jpcc.9b10279>

<https://doi.org/10.1021/acs.nanolett.9b02336>

POINT-BY-POINT REPLY TO REVIEWERS' COMMENTS

In the currently submitted manuscript – with minor changes in general - altered text (beyond simple, single rewording) is indicated in **red** and added text in *red/italics*.

The introduction part has significant changes. To ease the work of the Referees, we attached an additional, more detailed file on the introduction with relocated (with the relocation indicated by arrows) sentences indicated in red for the **target position**, in red/crossed for the **original position** in the previously submitted manuscript or removed and *red/italics* for newly introduces parts.

Supplement **S12** (Referee 2 / Q 3b) and **S13** (Referee 2 / Q 1) are entirely new and **S4** (Referee 2 / Q 3a) is strongly modified in response to Referee 2.

Reviewer #1 (Remarks to the Author):

The authors have fairly discussed mayor concerns addressed in the referee's reports and carried out additional measurements to support their interpretation. I think this work can be published now in Nature Communications.

Answer: We thank the reviewer for his/her support.

Reviewer #2 (Remarks to the Author):

I thank He-Yun Du et al. for their responses to my questions and updating their manuscript entitled “Nanoscale redox mapping at the MoS₂-liquid interface,” particularly in clarifying faradic, ion, and charge flow processes and why they modeled the SECM probe as a point source. I understand better their experimental setup and analysis. However, I still question the novelty of the work as it relates to spatially resolved measurements of mono-, bi-, and multilayer MoS₂ or other 2D materials, even at the nanoscale. In fact, the authors cite a paper (Ref. 36: MR Nellist et al. Nanotechnology 28, 095711 (2017)) demonstrating nanoscale redox mapping of 2D materials with the same commercial system (Bruker PeakForce AFM-SECM) used in the manuscript, and the sentence “The nanoelectrode probe is coated with dielectric materials and has an exposed conical Pt tip apex of ~200 nm in height and of ~25 nm in end-tip radius.” (line 62) is word-for-word the same as that in the abstract of that reference.

Answer: The very important technical work by Nellist et al. (Ref.36) indicates the potential which might be achievable in terms of resolution with tailored probes.

In our work, we do make the next step and demonstrate that our physical access is significantly enhanced with unprecedented results obtained by the combination of various techniques. The relevance of combining different methods is critical for advancements – and, in line with Referee 3, now more clearly elaborated in the introduction.

In their rebuttal, the authors mention “the evolution of existing techniques and the combination with a broad range of other methods” and that “TMDCs serves here as an example with new insight obtained – but our approach is not limited to the physics of TMDCs,” but the manuscript, as written, does not clearly describe in more detail how these insights can be expanded to other applications or materials. Given the body of existing work on MoS₂ electrochemistry and band structure (e.g., Choi, S et al. 2014; Trainer, D. J. et al. 2017; Velický, M. et al. 2016; Velicky and Toth 2017 cited in the manuscript) and the focus on demonstrating the use of scanning probe systems to characterize MoS₂, this manuscript may be better suited for a more technically oriented journal.

Answer: Velicky and Toth 2017 (Review article) “Conclusions: In contrast, the electrochemistry of graphene and other 2D materials is still in its “infancy”. There is no denying of the fact that rapid advances in these applications are invaluable but it should not be done entirely at the expense of the basic scientific understanding, which will, in a long term, lead to much smarter and more efficient ways of harnessing 2D materials’ potential. “

As Velicky in their excellent review appropriately summarized: Amazing progress has been achieved on 2D materials based on studies of specific physical properties: Trainer et al. on valence band shift with STM, Choi et al. on WPFM/WF, Velicky et al. 2016 on electron transport etc. However, our understanding remains fragmented as stressed in the very recent review paper.

This is, where our work goes beyond current boundaries – enhancing the resolution, enhancing our understanding of the chosen system as well as even of the imaging process (see Q25.2 of

the 1st Referee Report – which is one important technical but not the main objective of the manuscript). This is also now more accentuated in the revised introduction.

These are the specific comments and concerns I have after reading the authors' rebuttal:

(1) I appreciate the authors' diligence in reading and summarizing literature on heterogeneous electron transfer (HET) rates of Fc and agree with their caution in comparing values obtained under different experimental conditions (Q21.2). However, I think that putting their results (*how apparent HET and band gap relate to the number of layers and flake size*) in context with reported values is important.

Answer: Thanks to the reviewer's suggestion, we have modified and extended the table, which is now added as supplement S13.

Band gap (eV)	ML	CBM/VBM*	BL	CBM/VBM*	$\Delta E_{BL-ML}^{conduction\ band}$	ΔE_{BL-ML}^{gap}	Reference	Technique
	2.41	0.49/-1.92	2.21	0.38/-1.83	-0.11	-0.20	us	STS
2.40	0.45/-1.95	2.10	0.40/-1.70	-0.05	-0.30	Ref.29		
2.15	0.31/-1.84					Ref.31		
2.06	0.27/-1.79	1.89	0.27/-1.62	0	-0.17	Ref.30		
2.00	0.75/-1.25	1.75	0.50/-1.25	-0.25	-0.25	S ³		
1.98		1.8			-0.18	S ⁴	HREELS	
1.88		1.59			-0.29	Ref.19	Photoluminescence	
2.11(sapphire)	0.35/-1.76					Ref.3	ARPES/ARIPES	
1.90(Au)	0.6/-1.30							
Work function (eV)	ML		BL		$\Delta E_{BL-ML}^{work\ function}$	Reference	Technique	
	5.14		5.19		0.05	us (CVD)	KPFM	
4.93		4.88		-0.05	Ref.27(CVD)			
4.49		4.54		0.05	Ref.24(Exfoliate)			
5.15		5.25		0.1	Ref.23 (Exfoliate)			
4.64		4.69		0.05	S ⁵ (Exfoliate)			
4.36(ambient)					Ref.26			
4.04(UHV)					Ref.26			
4.93(SiO ₂ /Si)					Ref. 27			
5.10(Au)					Ref. 27			

We have revised / extended the table, which was previously only part of the answer to the Referee 2, and is added as Supplement S13. We agree that the collection might be interesting for a less specialized audience - although the principle behavior of band gap energy reduction with thickness is a commonly known phenomenon of many low dimensional systems. Therefore, we see no benefit in adding the values to the main article.

Absolute values given in different references for the conduction and valence band as well as the work function show some ambiguities (as systematic errors are methods related), however, the decrease of the band gap, the downward shift of the conduction band and the increase of the work function (highlighted in yellow) – which are not affected by such systematic errors under identical experimental conditions – as relevant for our analysis are in agreement with previous reports.

(2) Although I appreciate the addition of the photoelectrochemical measurements (Fig. 5), I don't see a clear correlation between the photoelectrochemical water-splitting at TMDCs mentioned throughout the manuscript and the measurements performed in organic solvents. I would have liked to see an explanation as to why organic supporting electrolyte was used (e.g., organic electrolytes were used instead of aqueous ones due to delamination of the MoS₂ flakes). I also would have liked to know how the band alignment in organic solvent would compare with that in aqueous solution (see for example, LF Schneerneyer and MS Wrighton "Flat-Band Potential of n-Type Semiconducting Molybdenum Disulfide by Cyclic Voltammetry of Two-Electron Reductants: Interface Energetics and the Sustained Photooxidation of Chloride" J. Am. Chem. Soc. 1979, 101, 6496)

Answer: We can't follow the referee in his attempt to see our results only in the singularity of water splitting.

- *We do not consider mentioning twice water-splitting inside the introduction as "throughout the manuscript" – as it is explicitly driven by the underlying electronic structure which is addressed in our manuscript.*
- *Fig. 5 is not discussed within the framework of water splitting – but in terms of the underlying physical system, here specifically to the band structure.*
- *Organic versus aqueous solution Is just another, but certainly interesting question, which is not the target of our current work. We see it as an interesting motivation for the future, but beyond the scope of this manuscript.*

(3) I appreciate the authors clarifying the electrochemical measurements by adding more details to the figure captions as well as correcting units and electrochemical details (e.g., supporting electrolyte). However, I stand by my original assessment that the interpretation and discussion of the electrochemical results are ambiguous:

Answer: As far as we see, all raised questions are clearly answered – some reported findings might be unexpected (see Q25.2 of previous report) - we give full access to all underlying details so that artefacts etc. can be unambiguously ruled out – and none (as far as we see) contradicts any previous knowledge but grant an unprecedented and new insight. If something would contradict, we would certainly (after deep self-questioning) not hesitate to respectively highlight such a finding.

(a) Regarding Q18.2, I still don't understand why the potential of Fc versus the SHE (standard hydrogen electrode) is the same that versus the Ag quasi-reference electrodes in Fig. S4c and how the authors arrived at their values. I would expect these values to be different, although I agree that 0 V vs. SHE is comparable to -4.44 eV for vacuum. The authors mention using both the (SHE) and Ag quasi-reference electrode (lines 100 and 107 in supporting information) as well as a "nonaqueous" silver-silver chloride reference in their response. The SHE is difficult to use in practice, and I wonder if they used a silver-silver ion electrode, which is more commonly used in organic electrolytes, instead of silver-silver chloride.

Answer:

- *We apologize for the confusion to the reviewer; it is indeed a nonaqueous silver-silver ion reference electrode that we used (now corrected in the revised version)*

- *We have clarified in Supplement S4 regarding the details on potential conversion.*

(b) In figure 5, I'm not sure why the current for the "off" photoelectrochemical measurements (280 to 300 pA) is much higher than that in the SECM images (ca. 1-15 pA). I would expect these values to be similar if the probe-MoS₂ distance is similar for all of the measurements.

Answer: Before answering the specific question, we note:

- *Purpose of Fig. 5c is qualitative proof-of-concept (validity unaffected by the Referee's questioning).*
- *Fig. 5c had a (now corrected) scaling error*
- *We kept Fig. 5c under a) correction of the scaling error and b) indication of the approximate background current*
- *Supplement S12 gives a detailed insight into additional experimental data from the same experimental run, adding a qualitative level but also illustrating the complications, which let us choose not to include a quantitative statement into the main manuscript.*

The observation by the Referee is correct and the values differ for a good reason significantly as the background current is not removed for Fig. 5c. We added Supplement S12, where we stepwise discuss a) data processing, b) the impact of potential artefacts and c) time-dependent data evolution for background information and a quantitative access see also answer to Question 4.

(c) The authors answered my question about using the word electronegativity to describe the reducing potential of redox-active species (e.g., lines 276 and 295) in their written response (Q38.2) but did not update the corresponding manuscript text.

Answer: We apologize, text changed.

(4) I appreciate the authors including CVs and approach curves of the SECM probe in Figure S5 (Q7.2). These additions help me better understand the experimental conditions. However, the probe was operated at 0.6 V, where the current is still changing (in other words, it is potential-dependent), and the current value is obviously not stable or reproducible at 0.6 V, as the probe current at 0.6 V was different for all three probes. I would have expected the probe to be poised at a potential (e.g., 0.7 V) where the current is at steady state (independent of potential) as well as more consistent between probes so that any change in current can be attributed to a change in redox activity of the substrate and not due to factors such as a drift in potential of the silver quasi-reference electrode.

Answer:

On the expense of higher current strength, the lifetime of probes reduces with higher potential, as experimentally observed – which is highly relevant for long-time stability in mapping and related aging effects. We therefore conducted experiments at a slightly lower potential 0.6V, with the drifts (potential, aging, tip changes etc.) treated by appropriate background handling. In test

experiments, we found no indication, that the reduced potential has any impact on the quality of made statements.

For illustration - not added as supplement and only for refereeing - we show above the evolution of the background with time related to Fig. 5c / S12. The experiment, which is demonstrated in Fig. S12 was redone on the very same surface with the same MoS₂ island present but at different potential settings. The above figure shows the cross-sections (which correspond the blue line in S12) over the insulating substrate in the slow-scanning direction with filtering but without background subtraction for 0.6 (black), 0.65 (red) and 0.75V (blue) versus Ag wire and the absolute current stated in the respective color on the left. The horizontal axis indicates the position (comparable to S12) and its respective conversion into time units (s).

With the change of the potential, (from black to blue), we see in the experimental data an increase of the absolute value originating from the substrate. We suspect - as not further addressed - that this reflects aging as experiments were conducted after each other and an unknown, i.e. not further investigated contribution of the potential. The absolute current, for the manuscript relevant current enhancement (values given in the Fig. above) is also increasing. From the experiments presented in Figs. 2 and 3, where we explicitly verified the impact of aging, we can unambiguously attribute here the signal increase on the MoS₂ to the changed potential - in accordance with the data from Cyclic voltammetry experiments. Parallel to the increased signal from MoS₂ also the modulation with light irradiation increases (values given in the Fig. above). At the same time, the data suggests (we did not explore further details as not relevant for the present study) that aging accelerates as given by ΔI_{drift} . We did not further elaborate on the specific details of potential variations.

We conclude:

- higher potential values tentatively increase the tip aging process
- we did not quantitatively further study the potential dependence
- qualitatively, the ratio of switching variations versus MoS₂ signal is rather unaffected, and can be equally used to demonstrate the proof-of-concept.

(5) For Q30.2, I am still unsure what a distant MoS₂ island is. If the island is another, isolated MoS₂ flake on Si/SiO₂ (an insulator), then I don't understand how charge (electrons) are being passed

through an insulator between the two flakes. If the island refers to two different areas on the same flake, then I can understand how the MoS₂ (a semiconductor) is able to act as a bipolar electrode.

Answer:

We deeply enjoy the depth of the Referee's detailed reading. The critically relevant information is inappropriately located at the end and should certainly appear at the respective place where needed.

- *We added in "We can **respectively** deduce a recycling radius to be on the range of ~200 nm [from here on added], **which reflects the obtained resolution at the step between the first and the second monolayer.**" – the information was previously only given in the summary.*
- *We added [in red] in: "**This leaves us with the unexpected finding of altered resolution from the first to the second monolayer. When the probe is located over an insulating surface like SiO₂, at a distant of up to 2 μm from a MoS₂ island (see Fig. 2h) the remaining feedback is driven by the charge flow only towards the more distant,**"*

Note: We certainly fully agree, that a charge transport through the insulator can be ruled out.

(6) For Q35.2, the authors may have misunderstood my question. I am not asking if they could combine all the measurement probes into one instrument but rather if they could do different measurements on the same type of sample, namely, measure MoS₂ on graphite (a conductive substrate) using the AFM-SECM and comparing those results to AFM-SECM imaging on MoS₂ on silicon oxide (an insulator) to see if the substrate (e.g., conductor vs. insulator) affects the observed electrochemical reactivity of the MoS₂.

Answer: This is an interesting question – which goes beyond the goal of our research as discussed in this manuscript. We certainly expect an impact on the electrochemical reactivity as the change of the substrate will affect for example the flow of the recharging current (see f. e. Chem. Commun., 2014, 50, 13117–13120) but also the entire electronic structure. We fully agree that such an approach will give a highly valuable, additional insight into this strongly growing field and see it as a suggestion for our future research.

Reviewer #3 (Remarks to the Author):

Comments to the Authors regarding to the resubmitted version of their revised version of “Nanoscale redox mapping at the MoS₂-liquid interface”

The authors revised their original work, and resubmitted an improved version, which attempts to solve the three Reviewers' comments. In my case, I see an improvement, and think that the manuscript had revised well, although it would improve more anytime. Regarding to my comments and question for the experimental and results parts are addressed, I am glad to see this. While one of my comment concerning the fundamental aspects of the current work is remained untouched, namely the substrate effect. I can understand that in the current COVID-19 situation, we should focus current results and utilise those as much as possible without trying to extend the existing content applying further experiments. But the Introduction part is still confusing and contains missing parts, pieces of works for me. Please, consider to revise more on the Introduction part to try to emphasize more the place and the role of your approach (the in-situ atomic force microscopy assisted SECM system). I mean we have knowledge of EC/PEC properties on MoS₂ and other TMDCs varying layer numbers, defect density, doping agent, surface treatments (passivation/activation), etc. Several research groups obtained these works using SECM, SECCM, and a microdroplet-based EC/PEC technique. I added some doi numbers of these papers below, supporting the Authors to find a way to emphasize the position of their work in the current view.

Answer:

We deeply thank the Referee for the comments and the reading advices, which we thoroughly studied and which opened a view on our own results within a broader context.

We respectively reworked/restructured our introduction, which previously handled separately and independently the physical properties, the application and our approach towards a better understanding of 2D / MoS₂ materials with the sharp link missing, which addresses the benefit of our, in the end, multi-methods approach. We intensively elaborated on the missing link, as suggested by Referee 3, which in parallel also tackles the criticism of Referee 2.

The revised version indicates the introduced changes of sentence arrangement with a newly introduced paragraph to stress the importance of exploring intertwined fundamental physical properties together, each individually contributing and together promising an over-performance in potential devices.

Introduction

Two-dimensional (2D) crystalline materials are consisting of a single layer of atoms and expected to have a significant impact on a large variety of applications^{1,2}. For transition metal dichalcogenide (TMDC) monolayers are direct semiconductors, of which a common representative is molybdenum disulfide (MoS₂). Monolayer MoS₂ has a device relevant direct bandgap of ~2 eV³ and its electron concentration at the surface than in bulk is enhanced by more than three orders of magnitude⁴. Their stacking variability and the dependence of physical properties with thickness are attractive for property fine-tuning⁵. TMDCs are often combined with other 2D layered materials like graphene or hexagonal boron nitride for device application such as transistors, solar cells, water splitting⁶, and sensors. The preparation of respective, high quality material on various substrates is established by means of chemical vapor deposition (CVD). The selection of growth parameters and precursors gives a precise control of its morphology, i.e. flake sizes and layer thicknesses⁷.

The well-defined band structure of MoS₂ flakes with fixed layer number is ideal to explore the localized electrochemistry of 2D materials as relevant for heterogeneous electron transfer (HET)⁸, photoelectrochemistry⁹, batteries/capacitors etc. Scanning electrochemical microscopy (SECM) is an established method¹⁰ to locally probe catalytic properties¹¹ as demonstrated for graphene oxide¹² and MoS₂ flakes in feedback mode¹³ and biased mode¹⁴. Whereas the device relevant absolute catalytic activity not only reflects the local catalytic property but its entanglement¹⁵⁻¹⁸ with the electronic structure, transport properties, geometrical structure, locally and in interaction with the entire environment and requires a multi-methods approach. The electronic structure of pristine ultrathin crystals of MoS₂ is intensively discussed in simulation and experiments¹⁹⁻²¹ with ambiguities related to growth conditions and measurement environments. ~~The: the work function is accessible~~ by Kelvin probe force microscopy (KPFM)²²⁻²⁷ and I-V curves in devices²⁸ ~~with results ranging from~~ ($\Phi = 4.49-5.15$ eV. ~~The~~), the energetic position of the conduction/valance band edge by scanning tunneling spectroscopy (STS)^{29,30} and X-ray photoelectron spectroscopy (XPS)^{31,32}, the optical band gap by photoluminescence spectroscopy (PL). ~~Monolayer MoS₂ has a device relevant direct bandgap of 1.8-2 eV and the electron concentration of MoS₂ is nearly four orders of magnitude higher at the surface than in bulk, which can be further fine-tuned by atomic vacancy states²¹.~~ The effects of band alignment at interfaces is so far mainly discussed for solid state TMDC based heterostructures^{31,33,34} ~~junctions~~ and semiconductor/liquid interfaces in general³⁵⁻³⁸. ~~Scanning electrochemical microscopy (SECM), also used here, is an established method to locally probe catalytic properties as demonstrated for graphene oxide and~~ with few reports on the role of the redox mediator-containing electrolyte on the interfacial band offset in MoS₂^{14,18} flakes in feedback mode and biased mode.

The research of recent years established thin film MoS₂ as a notably potent material for applications with a deep impact of edge and vacancy states^{39,40}, thickness⁴¹, domain sizes^{15,16}, electron transfer kinetics¹⁸ etc. on device performances, individually identified, although highly intertwined^{8,9,15,16,40,41}. The route towards highly efficient materials is the study and discussion of application relevant properties in conjunction with the full bandwidth of physical properties in different environments and their dependencies among each other^{17,18}. This is adressed here for MoS₂ and can be equivalently expected to be relevant for 2D materials in general, whenever dimensionality effects modulate electronic and chemical properties.

In SECM, when an AFM (atomic force microscopy)⁴² controlled probe approaches the un-biased semiconductor surface (feedback mode), a charge-flow through the local probe is established and gives access to the local electro-

chemical activity⁴³. ~~This provides a route to access directly the heterogeneous electron transfer (HET) between a 2D TMDC and a~~ The electrochemical response current, as detected by the AFM-SECM probe, maps the localized surface electrochemical activity between the 2D material and the liquid interface. Conventionally, SECM is performed in the lift mode, i.e. with the current collecting probe placed at a large distance of typically more than 100 nm above the reactive site. The lateral resolution depends on the probe radius and the distance from the surface and high-resolution imaging of various substrates is reported for ~~ultra-microelectrode probes. A further resolution enhancement can be expected when the probe is brought into closer proximity of the surface with the field gradient increased and respectively a focusing of the charge flow achieved. Accordingly, it remains challenging to measure and understand the interfacial band offset between MoS₂ and the redox mediator containing electrolyte.~~ nanoelectrode probes.

Here, we apply *in-situ* AFM assisted SECM (AFM-SECM) to layered MoS₂ with nanoelectrode probes⁴². With the probe operated in the AFM mode in close nm proximity, combining AFM (with observations on the scale of nm in x-y direction and atomic resolution in z direction) and SECM, with a minimum resolution of ~200 nm, to characterize in parallel 2D MoS₂ flakes, topological and electrochemical information in the near field, ~~with the probe operated in the AFM mode in close nm proximity. The nanoelectrode probe is coated with dielectric materials and has an exposed conical Pt tip apex of ~200 nm in height and of ~25 nm in end tip radius. Combining AFM and SECM~~ permits the deconvolution of electronic and topographical information and a more precise evaluation of the electrochemical activity. ~~AFM observations on the scale of nm in x-y direction and atomic resolution in z direction are given, with a min resolution of ~200nm observed in SECM maps. The electrochemical response current from MoS₂ monolayer is detected by the AFM-SECM probe and the localized surface electrochemical activity based on the HET behavior between 2D material and liquid interface accessed. Within~~ In combination with KPFM and STM/STS experiments and within the framework of the electron transfer at the solid-liquid interface, the HET behavior in dependence of size and layer numbers of the MoS₂ flake/liquid junction with Fc/Fc⁺ as a redox probe is revealed and discussed. ~~Thereby, the central relevance of the solvent mediator/electrolyte on the band alignment is revealed and further demonstrated by light interaction experiments.~~ identified and converted into an experimentally observable photocurrent variation.

methods section

Reviewers' Comments:

Reviewer #2:

Remarks to the Author:

The authors have sufficiently answered my questions and responded to my comments. The relationship between the presented results and previous literature is clearer as well as the analysis of the electrochemical data, as demonstrated by revisions in the main text and updated tables and figures in the supporting information.

Reviewer #3:

Remarks to the Author:

Thank you for the work of Authors, they replied to my questions and updated their manuscript, I see the improvement of this paper.